# ABCC4 impairs the clearance of plasma LDL cholesterol through suppressing LDLR expression in the liver
Jiaxin Chen[1,2,6], Hui Huang[2,6], Chi Chen[2,6], Guofang Xia[1,6], Hao Huang[2], Yan Xiong[2], Peng Luo[2], Yu Chen[2], Jinsong Li[2], Liang Wen[2], Lu Li [2], Jing Lin[2], Guangre Xu[3], Chenzhang Ji[4], Wenjie Tian[2], Jin Zhou[5] ✉, Peng Wei[1] ✉, Chengxing Shen[1] ✉ & Xiaoqing Wang [1,2] ✉

Low expression level of low-density lipoprotein receptor (LDLR) in hepatocytes leads to hypercholesterolemia and eventually contributes to atherosclerotic cardiovascular disease (ASCVD). Here, we report that inhibition of hepatocyte ABCC4, identified as a top hit from large-scale CRISPR/Cas9 screens, significantly increases hepatic LDLR abundance and enhances LDL cholesterol clearance. As a hepatic transporter for cAMP efflux, ABCC4 silencing alters its intracellular distribution and activates the downstream Epac2/Rap1a signaling pathway, which ultimately blocks PCSK9 protein expression, thereby preventing lysosomal degradation of LDLR. Furthermore, in both male mice and cell models, we demonstrate that liver-specific disruption and pharmacological inhibition of ABCC4 elevate hepatic plasma membrane LDLR levels and reduce plasma LDL cholesterol through ABCC4-cAMP-PCSK9 pathway. Collectively, our genome-wide CRISPR screening offers a valuable resource for identifying LDLR modifiers, providing potential insights for therapeutic strategies in hypercholesterolemia and atherosclerosis.

Low-density lipoprotein cholesterol (LDL-C) is a predominant form of lipids, which can be a driving force for the occurrence and progression of atherosclerotic events[1,2]. Aberrant accumulation of oxidized LDL particles within the coronary artery wall initiates monocyte recruitment and macrophage differentiation, contributing to a continuous process of inflammation[3–5]. Also, LDL-C is a key driver of atherosclerosis by stimulating smooth muscle cell (SMC) proliferation and progressive luminal narrowing, ultimately contributing to plaque expansion[6,7]. Prolonged exposure to elevated LDL-C levels predisposes individuals to coronary artery disease (CAD) and cerebrovascular disease (CVD), which is widely recognized as a major contributor to the overall risk.

In recent years, there has been a growing appreciation for the benefit of lowering serum LDL-C levels to reduce the incidence of atherosclerotic disease[8]. Proprotein convertase subtilisin/kexin type 9 (PCSK9), secreted by the liver, functions as a competitive binder to low-density lipoprotein receptor (LDLR), promoting LDLR lysosomal degradation and preventing its recycling back to the cell surface. The development of PCSK9 inhibitors (PCSK9i) has manifested their pivotal roles in lipid-lowing therapies by preventing the degradation of LDLR in hepatocytes[9,10]. Several approaches to PCSK9 blockade have been implemented for intervening hyperlipidemia in clinical practice. Repatha, a marketed PCSK9 monoclonal antibody, has demonstrated the ability to rapidly decrease LDL-C levels by 50–70% and lower the risk of myocardial infarction or stroke in multiple completed and ongoing clinical trials[11,12]. Besides PCSK9 antibodies, current clinical therapies including statins[13,14] and NPC1L1 (NPC1 like intracellular cholesterol transporter 1) inhibitors[15] enhance LDLR expression in hepatocytes, thereby promoting serum LDL-C clearance. Moreover, the inducible degrader of LDLR (IDOL), an E3 ubiquitin ligase that mediates the ubiquitination of hepatic LDLR[16,17], has emerged as a promising therapeutic target for cardiovascular disease, similar to PCSK9. Meanwhile, sterol regulatory element-binding protein 2 (SREBP2) promotes LDLR synthesis by binding to its promoter region[18,19] at the transcriptional level, thereby maintaining normal cellular cholesterol contents.

[1]Department of Cardiology, Shanghai Sixth People's Hospital Affiliated to Shanghai Jiao Tong University School of Medicine, Shanghai, China. [2]Institute of Cardiovascular Diseases & Department of Cardiology, Sichuan Provincial People's Hospital, School of Medicine, University of Electronic Science and Technology of China, Chengdu, China. [3]Sichuan Provincial People's Hospital, School of Medicine, University of Electronic Science and Technology of China, Chengdu, China. [4]Shanghai Biocam Pharmaceuticals Co. Ltd., Yangpu Shanghai, China. [5]Division of Liver Surgery, Department of General Surgery, West China Hospital, Sichuan University, Chengdu, China. [6]These authors contributed equally: Jiaxin Chen, Hui Huang, Chi Chen, Guofang Xia. ✉e-mail: zhoujin096@scu.edu.cn; pengwei_1983@sjtu.edu.cn; shencx@sjtu.edu.cn; Xiaoqing_Wang@uestc.edu.cn

As a transmembrane glycoprotein, low-density lipoprotein receptor (LDLR) is abundantly expressed on the surface of hepatocytes, where it facilitates the uptake of LDL cholesterol and maintains the homeostasis of lipid metabolism[20,21]. Approximately 75% of circulating LDL particles are cleared by LDLR through receptor-mediated endocytosis[22,23]. Following internalization, the acidic environment of the endosome induces conformational changes in LDLR, allowing it to dissociate from LDL particles. LDLR is then recycled back to the hepatocyte surface, while LDL cholesterol is trafficked to lysosomes for degradation[24–27]. Over the past decade, functional genomic strategies have emerged as powerful tools to systematically identify novel regulatory genes involved in metabolic processes and cardiovascular diseases. Accordingly, we hypothesize that additional novel and more effective modulators of LDLR remain to be identified through CRISPR-Cas9 functional screening.

In this study, we performed genome-wide screenings by introducing CRISPR libraries containing single guide RNAs (sgRNAs) targeting over 18,000 protein-coding genes into the AML12 hepatocyte cell line. Using this approach, we systematically identified key modulators of LDLR protein expression. Notably, we discovered ABCC4 as a promising hit gene governing LDLR surface abundance. Functioning as a hepatic efflux transporter of cyclic AMP (cAMP), ABCC4 deficiency led to intracellular accumulation of cAMP, thereby activating the downstream sensor Epac2. Activated Epac2 subsequently stimulates the small GTPase Rap1 and further suppresses PCSK9 protein expression, thus enhancing LDLR surface availability by preventing its lysosomal degradation. In general, our findings identified a key regulator of LDLR expression in hepatocytes, which offered novel insights into potential therapeutic targets for hypercholesterolemia and lipid-related disorders.

## Results

### CRISPR screen identifies ABCC4 as a negative regulator of hepatic LDL receptor

To investigate and identify additional novel and more effective modulators of LDLR, we herein performed fluorescence-activated cell sorting (FACS)-based genome-wide CRISPR screens in the AML12 (Alpha Mouse Liver 12) hepatocyte cell line. First, we transduced AML12 cells with a mouse genome-wide CRISPR knockout library (comprised of two independent parts, M1 and M2, each with different sgRNA designs for targeting the same genes) from Xiaole Shirely Liu Lab[28]. After transduction, we expanded the cells for another 3 days following blasticidin selection. Subsequently, cells were subjected to FACS to isolate the top 10% of cells with the highest LDLR surface expression (LDLR^high subpopulation) (Fig. S1a). Frequencies of sgRNAs in this sorted population were compared to those in the unsorted population (Fig. 1a). There was a significant decrease in the number of detected sgRNAs in the LDLR^high subpopulation, indicating a strong selection bias (Table S1 and S2). Following that, we applied the MAGeCK (Model-based Analysis of Genome-wide CRISPR-Cas9 Knockout) computational algorithm[29,30] to analyze our FACS-based CRISPR screen data (Fig. 1b, c; Data S1 and S2). At the conclusion of the screens, bar plots illustrated the log_2 fold change (log_2 FC) of the top-scoring genes in the LDLR^high subpopulation relative to the unsorted controls (Fig. S1b and S1c). Notably, we discovered previously reported modulators (UROD[31], CHP1[31], FASN[32], TMEM251[33], CDC42[34,35], REPS2[36,37]), as well as several novel candidate negative regulators of LDLR. Based on two independent screen replicates (M1 and M2), we found 68 gene hits from the M1 library that were also enriched as significant hits in the independent M2 screen. These hits were subsequently selected for further analysis (Fig. 1d; Data S3 and S4). ABCC4, the top-ranking putative negative regulator of LDLR identified in our screens, is expressed in various tissues, including the liver, brain, kidney, colon, lung and placenta[38,39]. Subcellularly, ABCC4 is localized to the basolateral membrane of hepatocytes, also referred to as the sinusoidal surface, where it transports several conjugated substrates into the bloodstream[40–42]. Moreover, ABCC4 has been shown to augment the potential for platelet activation by facilitating cAMP efflux, which in turn intensifies cAMP-dependent signal transduction to the nucleus[43,44].

Furthermore, cAMP and its downstream signaling pathways are known to regulate PCSK9 homeostasis and LOX-1 expression[45,46]. In addition to cAMP substrate, ABCC4 transports prostaglandins like PGE2, out of the cell, where they can bind to surface receptors and influence LDL-C levels[47]. Together, these previous studies align with our screening results, suggesting that ABCC4 functions as a negative regulator of cell surface LDLR expression.

### ABCC4 depletion increases the cell surface LDLR level independent of its gene expression

After the CRISPR screens, ABCC4 emerged as the most promising regulator of surface LDLR expression in AML12 cells. ATP-binding cassette subfamily C member 4 (ABCC4) is a member of the superfamily of ATP-binding cassette (ABC) transporters, which mediates ATP-dependent efflux of a diverse range of endogenous and exogenous substrates. Multiple downstream signals or substrates of ABCC4 have been identified in mammals, including the classic second messengers cAMP and cGMP[48,49], the eicosanoids PGE1/PGE2[50], and the bile acids cholytaurine[51]. Both *human* and *mouse* ABCC4 proteins are constitutively expressed in various tissues, including the liver, where they play essential roles in physiological transport processes, drug efflux and detoxification across the basolateral membrane of hepatocytes into the bloodstream[39–41]. To investigate the role of ABCC4 in regulating LDLR abundance, we first performed CRISPR-mediated single-gene knockout of *Abcc4/ABCC4* using two different sgRNAs (Fig. 2a, b). We then assessed the impact of *Abcc4/ABCC4* disruption on LDLR expression levels in both *murine* and *human* hepatocyte cell lines (Fig. 2; Fig. S2). As expected, *Abcc4* knockout significantly increased cell-surface LDLR abundance without altering total LDLR protein levels (Fig. 2c–f; Fig. S2a and S2b). To determine whether this increase resulted from transcriptional changes, we compared *Ldlr/LDLR* mRNA levels between control knockout (Rosa26-KO for *murine* cells, AAVS1-KO for *human* cells) and *Abcc4/ABCC4*-KO cells. No significant differences were observed in *Ldlr* mRNA expression (Fig. S2c), suggesting a post-transcriptional regulatory mechanism. Next, to evaluate whether lipoprotein endocytosis is dependent on *Abcc4*-mediated regulation of cell surface LDLR abundance, we incubated control-KO and *Abcc4*-KO cells with DiI-LDL particles. By flow cytometry and immunofluorescence analyses, we found that *Abcc4* deficiency markedly enhanced LDL uptake (Fig. 2g–j). This regulatory effect was further validated in *human* hepatocyte cell line LO2, where similar increases in LDLR surface expression and LDL uptake were observed (Fig. 2k–o; Fig. S2d–S2g). Additionally, cholesterol is essential for cellular growth and maintenance while its uptake, synthesis and metabolism, is largely regulated by sterol regulatory element-binding protein 2 (SREBP2) and 3-hydroxy-3-methylglutaryl-CoA reductase (HMGCR)[52,53]. To assess whether *Abcc4* deficiency affects intracellular cholesterol levels, we measured cellular cholesterol contents in *Abcc4*-deficient AML12 cells using ELISA method and found that ABCC4 barely exerted effect on cellular cholesterol contents (Fig. S2h). Consistently, the loss of *Abcc4* did not influence SREBP2-dependent gene expression (*Srebp2* and *Hmgcr*), as evidenced by RT-qPCR analysis (Fig. S2i). As shown in Fig. 2g–j, *Abcc4* knockout in hepatocytes had significant effects on cell surface LDLR expression and LDL uptake, without effects on sterol sensing and cellular cholesterol contents (Figs. S2c, S2f, S2h and S2i), suggesting the involvement of compensatory pathways for cholesterol export. Since LDL internalized via LDLR is subsequently trafficked to lysosomes for degradation, we further evaluated the degradation of DiI-LDL (after 4 h)[54,55]. Knockout of *Abcc4* in AML12 cells promoted the uptake of DiI-LDL, as well as the degradation of DiI-LDL (Fig. S2j), which may account for the minimal impact of ABCC4 on sterol sensing and cholesterol synthesis (LDLR, SREBP2 and HMGCR). Additionally, hepatocytes harbor robust metabolic systems for converting cholesterol into bile acids or oxysterols, which likely mask the impact of increased cholesterol influx. Taken together, these data indicate that ABCC4 blockade enhances hepatic LDLR surface expression and lipoprotein uptake independent of LDLR gene transcription.

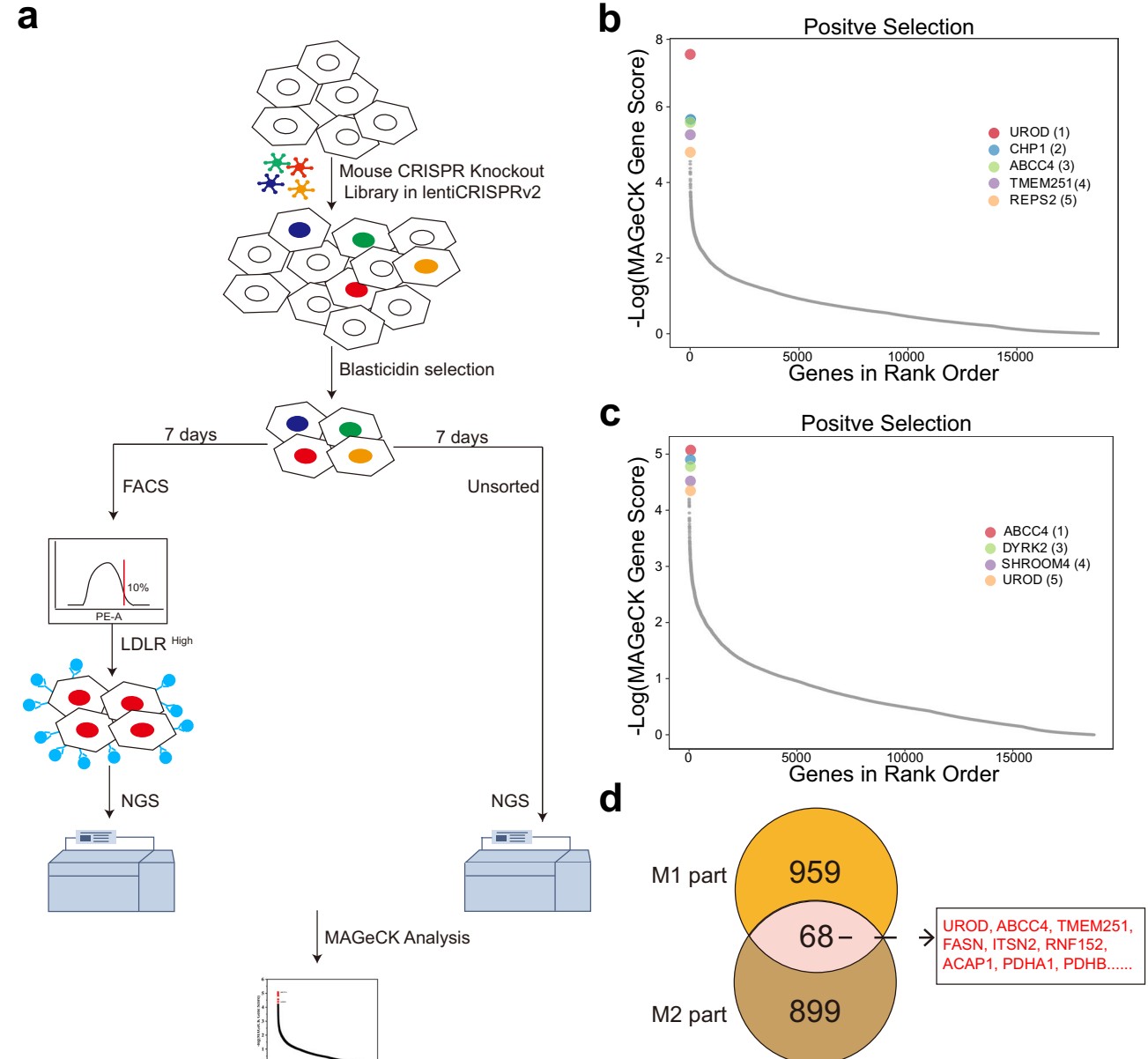

**Fig. 1 | CRISPR screen identifies ABCC4 as a negative regulator of hepatic LDL receptor. a** Schematic workflow of genome-scale CRISPR screening process. **b, c** MAGeCK gene enrichment scores comparing LDLR^high subpopulation and unsorted cells from two independent screens (M1 and M2). **d** Venn diagram showing genes identified in two independent biological replicates (M1 and M2).

## Liver-specific disruption of ABCC4 promotes hepatic plasma membrane LDLR protein expression and lowers plasma LDL cholesterol levels in vivo

Given that we observed the significant effects of ABCC4 blockade on regulation of LDLR availability in hepatocytes, this drives us to investigate whether ABCC4 plays a role in regulating plasma lipoprotein metabolism. To further assess the biological function of ABCC4 in vivo, we administered adeno-associated virus serotype 8 (AAV8) carrying an RNA interference (RNAi) construct targeting *Abcc4* (AAV8_Abcc4_RNAi) under the control of the liver-specific ApoE/hAAT promoter[56,57] to wild-type (WT) C57BL/6 male mice ($n = 6$, $2 \times 10^{11}$ viral genomes (VG) per animal) (Figs. 3a and S3a). Comparing to AAV8_GFP_RNAi controls, AAV8_Abcc4_RNAi administration in mice lowered hepatic *Abcc4* mRNA expression levels by about 44.9% (Fig. 3b) and protein levels by about 64.7% (Fig. 3c, d). Reduced hepatic ABCC4 level did not significantly affect body weight (Fig. S3b). But, it remarkably increased hepatic membrane LDLR protein expression and

accordingly decreased serum LDL-C levels, without significant changes in serum total cholesterol (TC) and triglycerides (TG) levels (Fig. 3e-i). This observation aligns with prior findings that LDLR abundance on hepatocyte membranes is inversely correlated with circulating LDL-C levels in both mice and humans[58]. As shown in vitro experiments, ABCC4 barely influenced intracellular cholesterol content (Fig. S2h). We therefore measured liver TC and TG levels in mice and concordantly illustrated that there were no significant alterations in AAV_Abcc4_RNAi group fed with a normal-chow diet (NCD) (Fig. S3c). These results reinforce the role of ABCC4 in regulating hepatic LDLR expression and plasma LDL cholesterol levels.

## Pharmacologic inhibition of ABCC4 by Ceefourin-1 facilitates plasma LDL cholesterol clearance by enhancing surface LDLR protein expression on hepatocytes

Previous studies have demonstrated that Ceefourin-1 is a highly specific inhibitor of ABCC4[59–62], effectively blocking the transport of its substrates

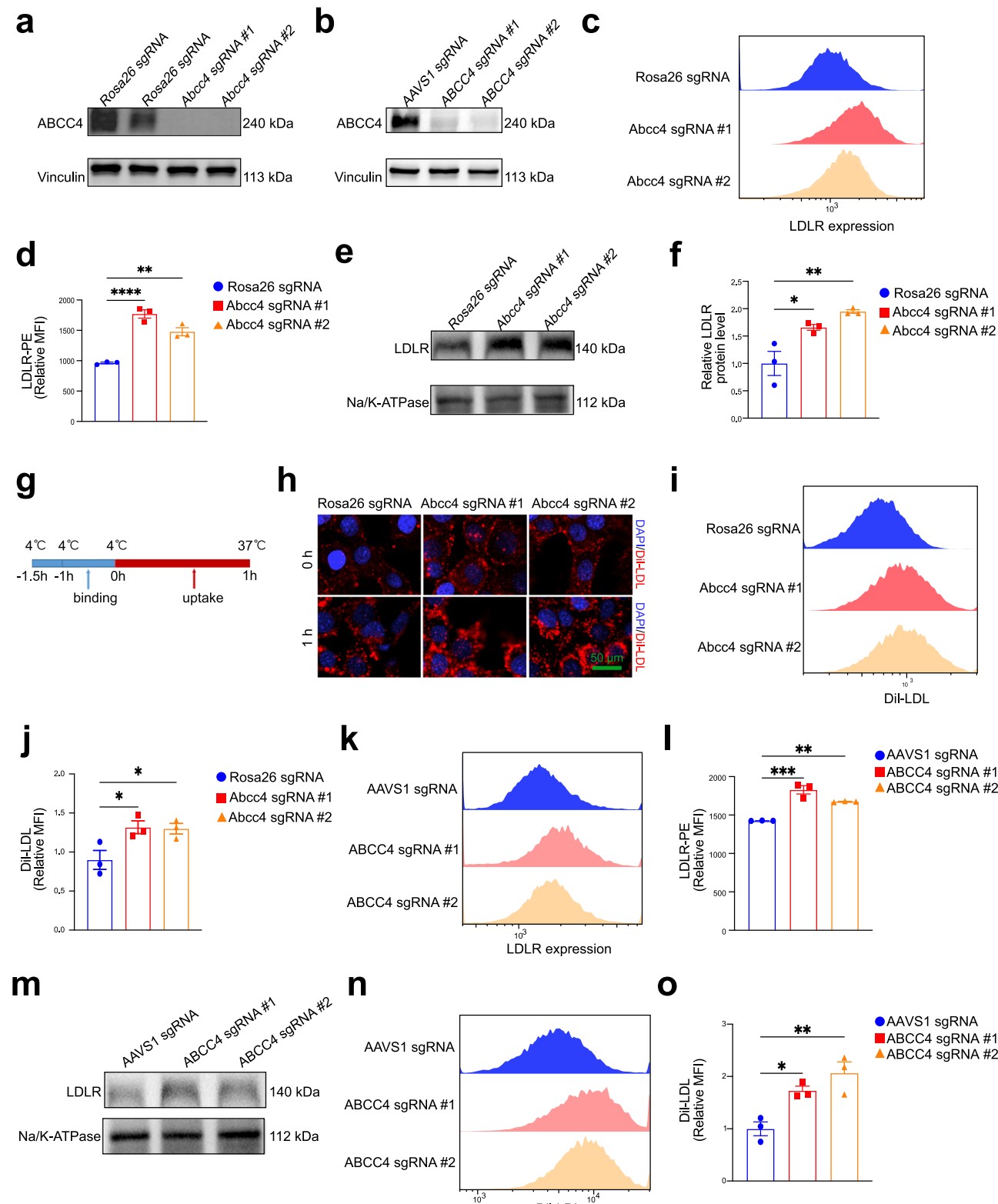

such as 6-mercaptopurine (6-MP), cAMP, D-luciferin, and estradiol-17β-glucuronide (E217βG), without exerting detectable effects on related transporters MRP1, MRP2, MRP3 or MRP5. Ceefourin-1 has been shown to inhibit cAMP efflux, thereby reducing platelet adhesion and thrombus formation[61]. To further explore the role of ABCC4 signaling in modulating LDLR expresion and LDL-C levels, we employed Ceefourin-1 to pharmacologically inhibit ABCC4 both in vitro and in vivo. In line with previous findings[63], Ceefourin-1 treatment exerted little effect on ABCC4 protein

expression levels (Fig. S4a and S4b). Nevertheless, Ceefourin-1 significantly potentiated cell surface LDLR abundance in AML12 cells by flow cytometry (Fig. 4a, b) and immunoblotting analyses (Fig. 4c, d). Thus, we further evaluated the effect of pharmacological inhibition of ABCC4 on the clearance of LDL-C in male mice fed with a normal-chow diet (NCD) or high-fat diet (HFD) (Fig. 4e). Administration of Ceefourin-1 (10 mg/kg) or vehicle (DMSO) for 4 weeks did not significantly alter body weight (Fig. 4f) or hepatic ABCC4 protein expression levels (Fig. S4c and S4d) in male mice fed

**Fig. 2 | ABCC4 depletion increases the cell surface LDLR level independent of its gene expression. a, b** CRISPR/Cas9–mediated knockout of Abcc4 in AML12 cells (**a**) and LO2 cells (**b**) using two independent sgRNAs. Immunoblotting analysis of ABCC4 protein expression and Vinculin in control-KO cells and *Abcc4*-KO cells (Rosa26 sgRNA, Abcc4 sgRNA #1, Abcc4 sgRNA #2). **b** Immunoblotting experiments of ABCC4 and Vinculin in control-KO cells and ABCC4-KO cells (AAVS1 sgRNA, ABCC4 sgRNA #1, #2). **c** Flow cytometry data showing that knockout of *Abcc4* increased the abundance of LDLR on the AML12 cell surface (Representative data from *n* = 3 independent experiments with similar results). **d** Plot showing the relative Mean Fluorescence Intensity (MFI) of PE-LDLR from three independent experiments. **e** Immunoblotting analysis of plasma membrane fractions demonstrating that *Abcc4* knockout dramatically increased the amount of LDLR on PM. **f** Quantification of band intensity of LDLR protein expression relative to Na/K-ATPase from three independent experiments. **g** Cultured AML12 cells were precooled to 4 °C for 30 min and incubated with Dil-LDL for binding at 4 °C. Then the cells were washed 3 times with ice-chilled PBS. The cells were substantially switched to 37 °C for uptake. **h** Representative immunofluorescence microscopy images showing that *Abcc4* knockout in AML12 cells had potentiated influence on DiI-LDL binding (0 h) and uptake (1 h). Blue: DAPI; Red: Dil-LDL. Scale bar: 50 µm. **i** Dil-LDL uptake assay implying that LDL uptake was significantly promoted in *Abcc4*-deficient cells by flow cytometry analysis (Representative data from *n* = 3 independent experiments with similar results). **j** The relative MFI of Dil-LDL quantification were from 3 independent experiments. **k** FC data showing that knockout of ABCC4 promotes the cell surface LDLR accessibility in LO2 cells (Representative data from *n* = 3 independent experiments with similar results). **l** The relative MFI of LDLR-PE quantification was from three independent experiments. **m** Immunoblotting analysis of LDLR quantification located on the plasma membrane fractions relative to Na/K-ATPase in LO2 cells. **n** Flow cytometry analysis showing that LDL uptake by LDLR was significantly promoted in LO2 cells lacking ABCC4 (Representative data from *n* = 3 independent experiments with similar results). **o** The relative MFI of Dil-LDL quantification were from three independent experiments. Statistical analysis was performed by a ordinary one-way ANOVA followed by Bonferroni's multiple comparison test in (**d**), (**f**), (**j**), (**l**), (**o**). *$P \leq 0.05$, **$P \leq 0.01$, ***$P \leq 0.001$, ****$P \leq 0.0001$, ns: no significance. Data are the mean ± SEM. Source data are provided as a Source Data file.

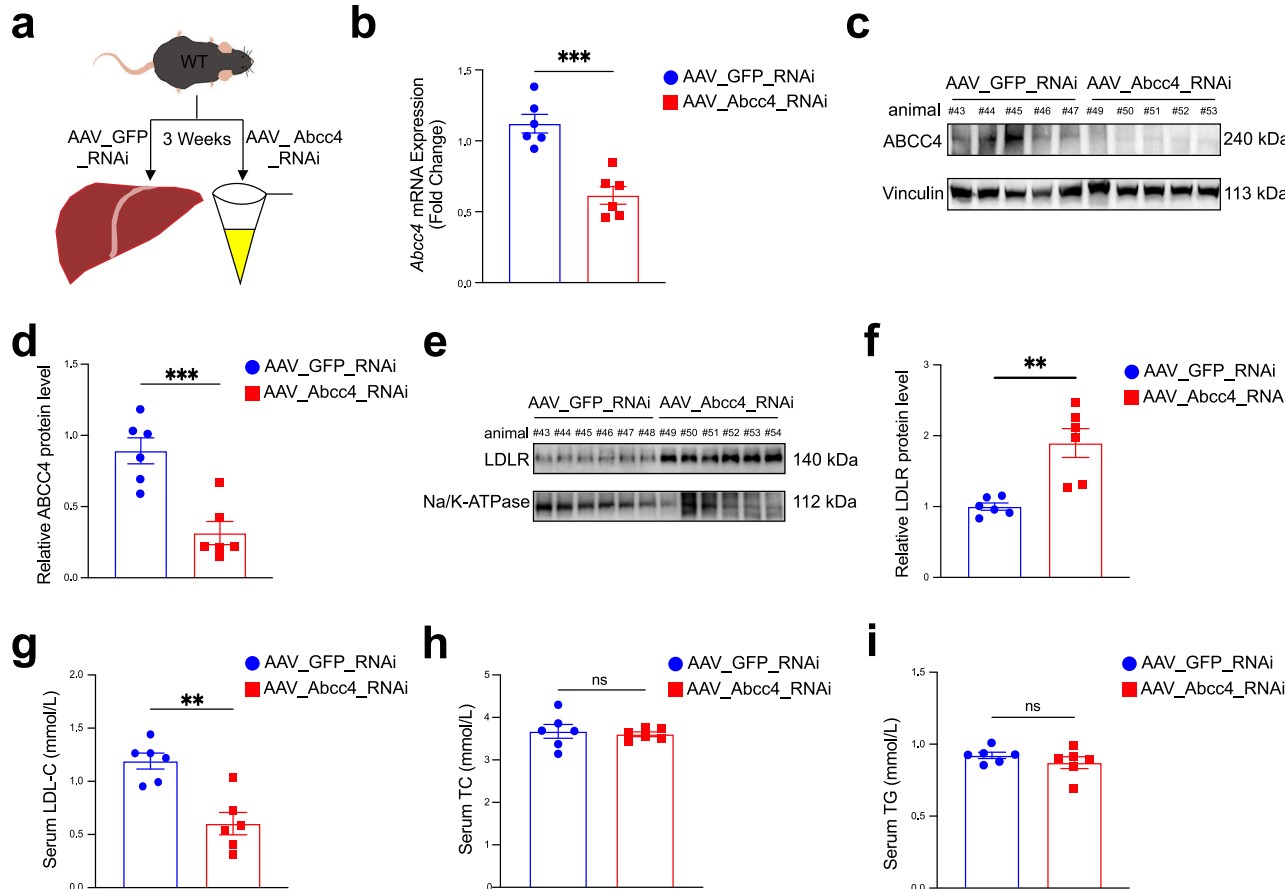

**Fig. 3 | Liver-specific disruption of ABCC4 promotes hepatic plasma membrane LDLR protein expression and lowers plasma LDL cholesterol levels in vivo. a** WT C57BL/6 male mice were treated with control AAV_GFP_RNAi or AAV_Abcc4_RNAi by tail vein injection ($2 \times 10^{11}$ viral genomes per mouse, *n* = 6 per group) for 3 weeks. **b** Hepatic *Abcc4* mRNA expression level in WT mice between two groups. **c** Representative immunoblotting data of hepatic ABCC4 protein expression level in WT mice between two groups. **d** Quantification of band intensity of ABCC4 protein level relative to Vinculin in liver tissue from two groups. **e** Representative immunoblotting data of membrane LDLR protein expression from liver tissue membrane fractions in mice. **f** Quantification of band intensity of membrane LDLR protein level relative to Na/K-ATPase in liver tissue from two groups. **g** Serum LDL-C level in WT mice between two groups. **h** Serum TC level in in WT mice between two groups. **i** Serum TG level in WT mice between two groups. Statistical analysis was performed by an unpaired two-tailed Student's t-test in (**b**), (**d**), (**f**), (**g**), (**h**), (**i**). **$P \leq 0.01$, ***$P \leq 0.001$, ns: no significance. Data are the mean ± SEM. Source data are provided as a Source Data file.

with either a normal chow diet (NCD) or high-fat diet (HFD). As expected, Ceefourin-1 treatment significantly reduced serum LDL-C levels by about 41.1% in NCD-fed mice and 42.2% in HFD-fed mice (Fig. 4g). In contrast, no significant changes were observed in serum TC or TG levels between treated and control groups (Fig. 4h, i). Interestingly, Ceefourin-1 treatment

markedly lowered liver TC and TG contents in HFD-fed mice (Fig. 4j, k). This was further supported by H&E and Oil Red O staining, which revealed a substantial reduction in hepatic lipid accumulation, particularly in the HFD group (Fig. 4l). The liver is a central player in the maintenance of glucose and lipid homeostasis, where hyperglycemia alongside with insulin

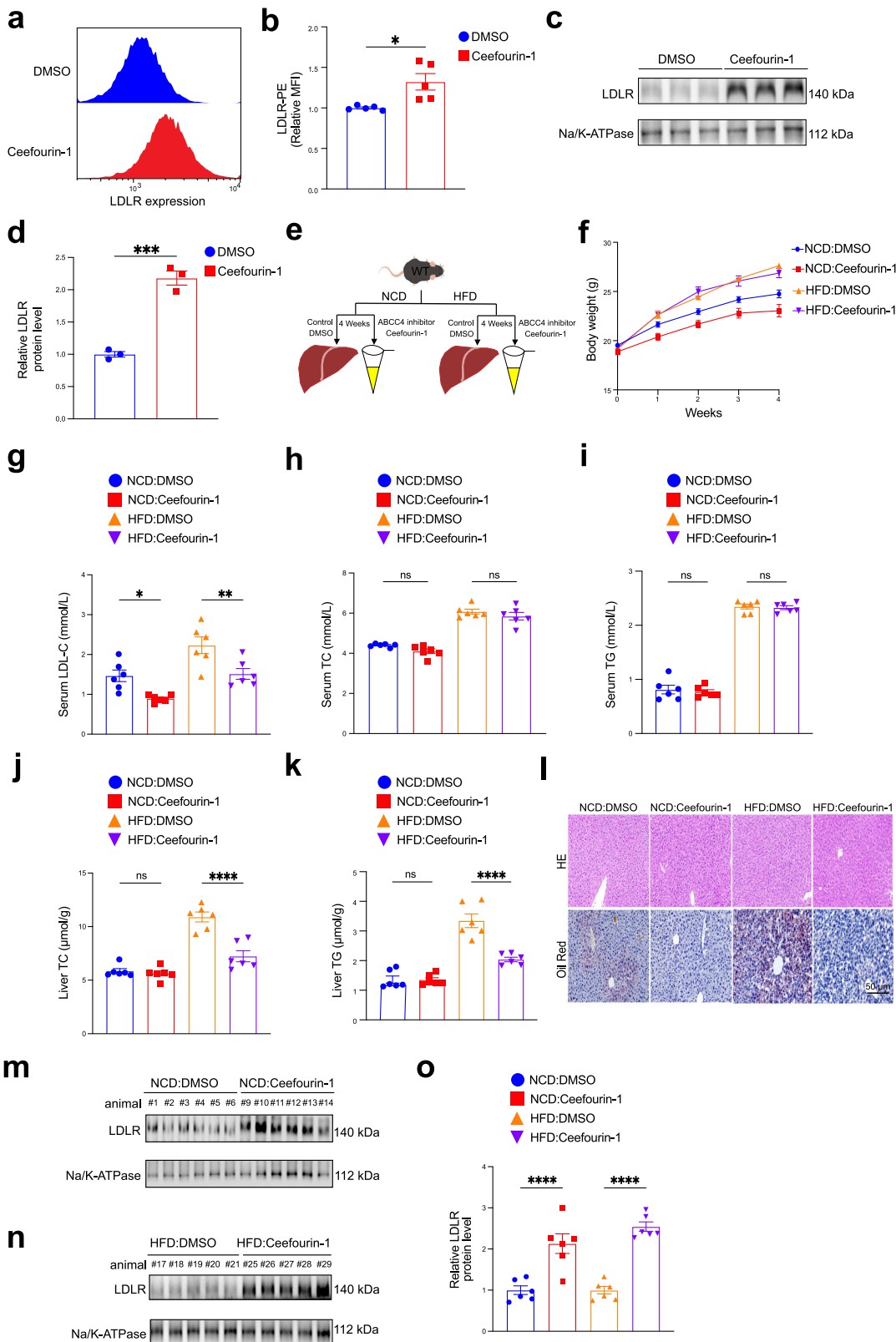

resistance are predominant manifestations of numerous metabolic abnormalities and diseases[64–67]. Hence, we hypothesized that Ceefourin-1 might modulate glucose metabolism and de novo lipogenesis, thus alleviating hepatic lipid accumulation. Then, we assessed fasting blood glucose and serum insulin levels, as well as glucose and insulin tolerance, in mice treated with Ceeforuin-1 or DMSO. As anticipated, HFD feeding significantly elevated fasting blood glucose and serum insulin levels compared with NCD-fed mice (Fig. S4e). Notably, Ceefourin-1 treatment improved glucose tolerance and insulin sensitivity in HFD-fed mice, as demonstrated by glucose tolerance tests (GTT) (Fig. S4f) and insulin tolerance tests (ITT) (Fig. S4g). These findings suggest that the reduction in hepatic cholesterol contents may be attributed, at least in part, to the

**Fig. 4 | Pharmacologic inhibition of ABCC4 by Ceefourin-1 facilitates plasma LDL cholesterol clearance by enhancing surface LDLR protein expression on hepatocytes. a** Flow cytometry data showing that ABCC4 inhibitor treatment potentiates surface LDLR availability (Representative data from $n = 5$ independent experiments with similar results). **b** Plot showing the relative Mean Fluorescence Intensity (MFI) of PE-LDLR from five independent experiments. **c** Immunoblotting experiments of the plasma membrane fractions demonstrating that ABCC4 inhibitor treatment up-regulates surface LDLR protein expression in AML12 cells. **d** Quantification of band intensity of surface LDLR protein expression relative to Na/K-ATPase in AML12 cells (Data from $n = 3$ independent experiments). **e** Wild-type male mice ($n = 6$ per group) were injected intraperitoneally three times a week with a dose of Ceefourin-1 (10 mg/kg) or vehicle control (DMSO and corn oil) for 4 weeks under normal-chow diet (NCD) or high-fat diet (HFD) conditions. **f** Plot displaying changes of body weight in mice intraperitoneally with vehicle control and ABCC4 inhibitor Ceefourin-1 under a NCD or HFD condition. **g** Serum LDL-C level in WT mice treated as in (**e**). **h** Serum TC level in WT mice treated as in (**e**). **i** Serum TG level in WT mice treated as in (**e**). **j** Liver TC level in WT mice treated as in (**e**). **k** Liver TG level in WT mice treated as in (**e**). **l** Hematoxylin and eosin (H&E) and Oil Red O staining analysis revealing that ABCC4 inhibitor treatment improved lipid accumulation, especially under a HFD condition. Scale bar: 50 μm. **m** Immunoblotting data of LDLR protein expression from liver plasma membrane fractions in NDC-fed mice treated with Ceefourin-1 or vehicle. **n** Representative immunoblotting data of LDLR expression from liver plasma membrane fractions in HFD-fed mice treated with Ceefourin-1 or vehicle. **o** Quantification of band intensity of LDLR protein level relative to Na/K-ATPase in liver plasma membrane fractions from the mice treated as in (**e**). Statistical analysis was performed by an unpaired two-tailed Student's t-test in (**b**), (**d**); a ordinary one-way ANOVA followed by Bonferroni's multiple comparison test in (**g**), (**h**), (**i**), (**j**), (**k**), (**o**). *$P \le 0.5$, **$P \le 0.01$, ***$P \le 0.001$, ****$P \le 0.0001$, ns: no significance. Data are the mean ± SEM. Source data are provided as a Source Data file.

suppression of glucose-driven lipogenesis rather than decreased cholesterol uptake alone. Furthermore, consistent with our in vitro findings, Ceefourin-1 administration significantly increased hepatic surface LDLR protein expression under both NCD and HFD conditions (Fig. 4m–o). Altogether, blocking ABCC4 function by a specific inhibitor exerts a protective effect on lipid metabolism by enhancing LDLR availability and mitigating hypercholesterolemia, highlighting ABCC4 as a promising therapeutic target.

Our findings demonstrate that both genetic ablation (Fig. 3) and pharmacological inhibition of hepatic ABCC4 (Fig. 4) effectively elevate hepatic LDLR protein levels and reduce plasma LDL-C levels by approximately 40% in mice. Notably, ABCC4 protein is highly conserved between *humans* and *mice*, with 100% sequence identity, suggesting functional conservation across species[39,41,42]. To further explore the association between ABCC4 and plasma LDL-C levels in larger populations, we analyzed publicly available datasets from the UK Biobank and the Global Lipid Genetics Consortium[68,69]. We identified additional variants at the *ABCC4* gene locus that were significantly associated with plasma LDL cholesterol levels (Fig. S5a). Moreover, analysis of public data from the STARNET (Stockholm-Tartu Atherosclerosis Reverse Networks Engineering Task) study[70] revealed that ABCC4 belongs to a liver gene co-expression network module significantly linked to coronary artery disease and other cardiometabolic outcomes (Fig. S5b).

## Altered intracellular distribution of cAMP by ABCC4 blocks hepatic PCSK9 expression

To confirm that the regulation of hepatic surface LDLR abundance by ABCC4 signaling is intrinsic to hepatocytes, we utilized the AML12 in vitro cell model. Transcriptomic profiling revealed a clear separation between *Abcc4* sgRNA (*Abcc4*-KO) and *Rosa26* sgRNA (control-KO) AML12 cell clusters (Fig. 5a), indicating distinct gene expression landscapes. Differential expression analysis identified 1,290 genes, with 940 (72.87%) significantly upregulated and 350 (27.13%) downregulated in Abcc4-KO cells ($P < 0.05$; fold change>2.0; Data S5). A circos heatmap of the top 43 DEGs illustrated that *Abcc4* deletion primarily resulted in gene upregulation (Fig. 5b). Moreover, an in-depth analysis of these DEGs revealed significant enrichment in pathways including the cAMP signaling pathway, Rap1 signaling pathway, cGMP-PKG signaling pathway, Phospholipase D signaling pathway, insulin signaling pathway, insulin resistance, Glucagon signaling pathway and PPAR signaling pathway, which were closely implicated in lipid metabolism, as supported by previous studies (Fig. 5c)[45,71]. Gene set enrichment analysis (GSEA) further confirmed that *Abcc4* deficiency simultaneously activated a broad spectrum of genes involved in the cAMP signaling pathway (Fig. 5d), Rap1 signaling pathway (Fig. 5e). Additionally, *Abcc4* knockout significantly altered expression patterns of genes related to lipid metabolism process and insulin signaling pathway (Fig. 5f). Consistently, comprehensive transcriptome analysis of AML12 cells treated with DMSO or Ceefourin-1 revealed strikingly consistent gene expression profiles

(Fig. S6). Aligned with our in vivo observations, pharmacological inhibition of ABCC4 notably affected insulin-related signaling pathways (Fig. S4g).

Obviously, knockout of *Abcc4* in AML12 cells effectively inhibited cAMP efflux and promoted activation of the intracellular cAMP signaling pathway, as revealed by RNA-seq analysis. These findings were further validated using ELISA, which demonstrated elevated intracellular and reduced extracellular cAMP levels in *Abcc4*-deficient cells (Fig. 5g). Similarly, Ceefourin-1 treatment altered cAMP distribution in AML12 cells, supporting the role of ABCC4 in cAMP transport (Fig. 5h). It is widely acknowledged that clinically approved therapies, such as statins and PCSK9 inhibitors, offer mechanistic insight into LDLR regulation by altering cholesterol metabolism and arresting LDLR lysosomal degradation. Building on our earlier findings that *Abcc4* deficiency in hepatocytes enhances LDL uptake and accelerates its lysosomal degradation, without affecting sterol sensing and cholesterol levels (Fig. 2g–j; Figs. S2c, S2f, S2h-S2j), we hypothesized that ABCC4 inhibition might offer complementary or synergistic benefits. Then, we investigated whether increased surface LDLR abundance observed with hepatic *Abcc4* deficiency was interrelated to PCSK9 regulation. As anticipated, we observed a corresponding decrease in both intracellular and secreted PCSK9 protein levels without significant changes in *Pcsk9* mRNA expression in hepatocytes lacking *Abcc4* (Fig. 5i–l). Importantly, Ceefourin-1 treatment recapitulated these effects on PCSK9 protein, with no significant changes in Pcsk9 transcript levels (Fig. 5m–p). In animal models, both liver-specific disruption and pharmacological inhibition of ABCC4 significantly resulted in decreased hepatic PCSK9 protein expressions and lower plasma PCSK9 levels compared to the control group (Fig. S7a-S7e). Given that *Abcc4* deficiency did not alter *Pcsk9* mRNA levels (Fig. 5l), yet resulted in reduced intracellular and secreted PCSK9 levels (Fig. 5i-k), these finding suggest that ABCC4 exerts a posttranscriptional regulatory effect on PCSK9 expression. Therefore, to determine whether ABCC4 influences PCSK9 protein stability, we treated cells with cycloheximide (30 min and 90 min) to block protein synthesis. This treatment (90 min) led to a marked reduction in PCSK9 protein levels in *Abcc4*-KO cells, suggesting increased degradation (Fig. 5q). To further explore whether PCSK9 mediates the effect of ABCC4 on LDLR, we incubated AML12 cells with recombinant human PCSK9 protein (rhPCSK9; 500 ng/ml). Remarkably, rhPCSK9 treatment completely abolished the effect of *Abcc4* silencing on LDLR surface expression and LDL uptake (Fig. 5r, s), implicating that ABCC4 plays a critical role in modulating PCSK9 levels and, consequently, influencing LDLR protein expressions and LDL-C clearance. Given previous reports linking glucagon signaling to PCSK9 regulation[45,71], our RNA-seq data also revealed enrichment of glucagon pathway genes in *Abcc4*-KO cells (Fig. 5c). In follow-up experiments, we indeed found that glucagon (Glu) treatment significantly reduced both intracellular and secreted PCSK9 levels, mimicking the effect of Ceefourin-1. Combined treatment with glucagon and Ceefourin-1 further suppressed PCSK9 levels, suggesting a synergistic effect and reinforcing the mechanistic role of

ABCC4 in PCSK9 regulation (Fig. S8a and S8b). Together, our findings establish that activation of the intracellular cAMP signaling pathway is essential for regulating PCSK9 and LDLR expression in *Abcc4*-deficiency hepatocytes. Targeting ABCC4, therefore, could be a promising therapeutic strategy for hypercholesterolemia and lipid-associated metabolic disorders.

## ABCC4-cAMP signaling regulating cell surface LDLR availability is dependent on essential roles of Epac2/Rap1a

It is well established that the intracellular effects of cAMP are primarily mediated by two classes of effectors: the classical protein kinase A (PKA)/cAMP-dependent protein kinase and the exchange protein directly activated by cAMP (EPAC)/cAMP-regulated guanine nucleotide exchange

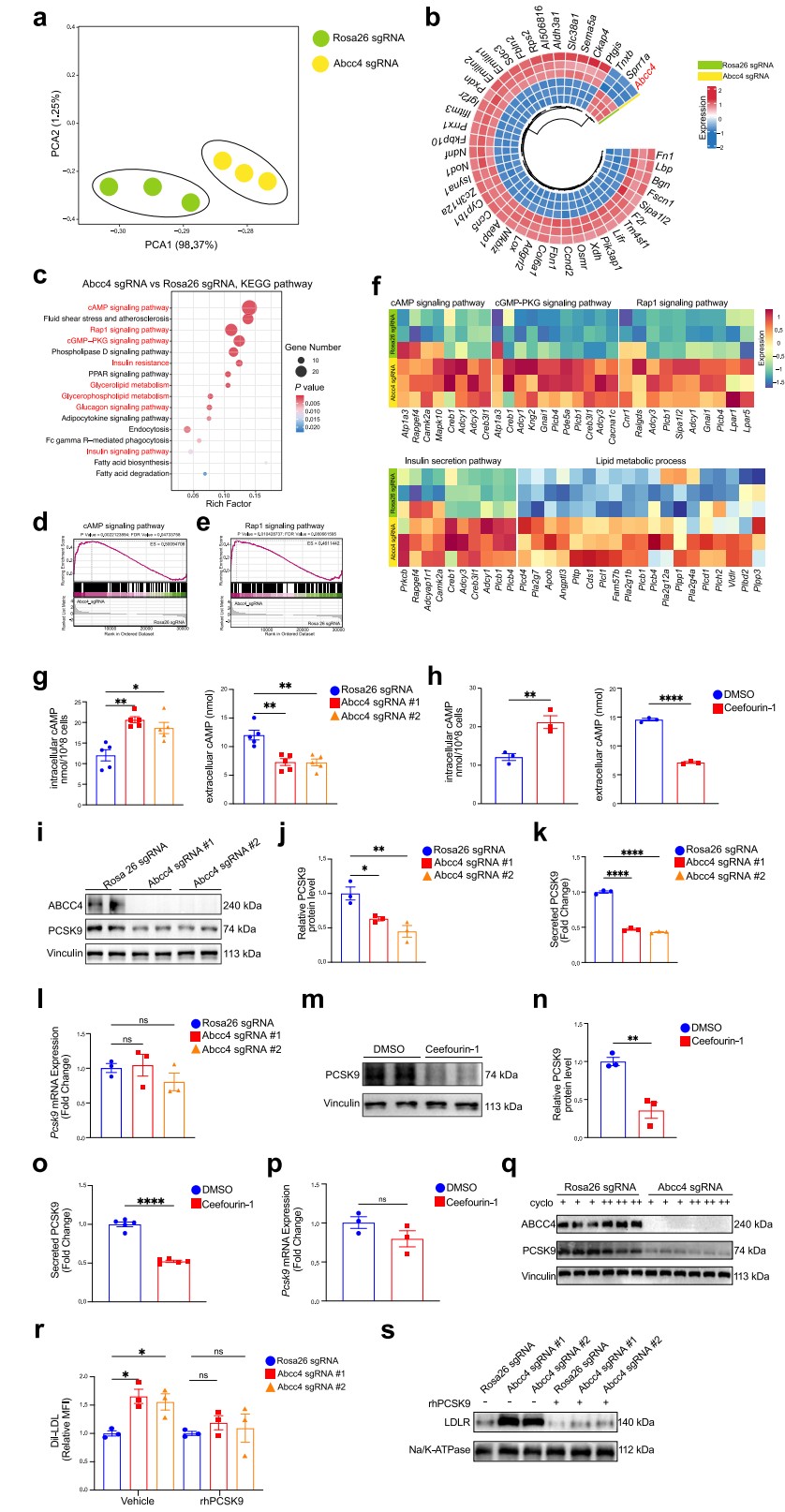

**Fig. 5 | Altered intracellular distribution of cAMP by ABCC4 blocks hepatic PCSK9 expression.**
**a** Principal component analysis showing distinct clustering of transcriptomes of the samples between *Abcc4*-knockout (Abcc4 sgRNA) and control-knockout (Rosa26 sgRNA) AML12 cells. **b** Hierarchically clustered circos heatmap of top 43 differentially expressed genes (DEGs) between two groups ($P < 0.05$; fold change>2.0). **c** KEGG pathway enrichment analysis of DEGs. **d, e** GSEA showing activated cAMP (**d**) and Rap1 (**e**) signaling pathways for DEGs between two groups. **f** Heatmap showing the enriched pathways related to lipid metabolism process, insulin secretion signaling, cAMP signaling, cGMP-PKG signaling, and Rap1 signaling for DEGs. **g** ELISA measurements showing increased intracellular and reduced extracellular cAMP levels in *Abcc4*-deficient cells. **h** ELISA measurements showing increased intracellular and reduced extracellular cAMP levels in AML 12 cell treated with the ABCC4 inhibitor Ceefourin-1. **i** Immunoblotting experiments showing PCSK9 protein levels in *Abcc4*-deficient cells. **j** Quantification of band intensity of PCSK9 protein expression relative to Vinculin in *Abcc4*-deficient cells from three independent experiments. **k** Secreted PCSK9 levels in *Abcc4*-deficient cells. **l** RT-qPCR results assessing *Pcsk9* mRNA level in *Abcc4*-deficient cells from three independent experiments. **m** Immunoblotting experiments showing PCSK9 protein level in AML12 cell treated with DMSO or Ceefourin-1. **n** Quantification of band intensity of PCSK9 protein level relative to Vinculin between two groups from three independent experiments. **o** Secreted PCSK9 levels in AML12 cells treated with DMSO or Ceefourin-1. **p** *Pcsk9* relative mRNA expression level by RT-qPCR between two groups from three independent experiments. **q** Immunoblotting analysis of PCSK9 protein levels in *Abcc4*-deficient cells treated with cycloheximide (cyclo: 4 µg/mL) for 30 min (+) and 90 min (++). **r** Flow cytometry analysis of Dil-LDL uptake assay (1 h) in *Abcc4*-deficient cells treated with Vehicle or rhPCSK9 protein. The relative MFI of Dil-LDL quantification were from three independent experiments. **s** Immunoblotting analysis of LDLR expression from the plasma membrane fractions in *Abcc4*-deficient cells. Statistical analysis was performed by a Welch ANOVA test followed by a post hoc analysis using the Tamhane T2 method in (**g**); an unpaired two-tailed Student's t-test in (**h**), (**n**), (**o**), (**p**); a ordinary one-way ANOVA followed by Bonferroni's multiple comparison test in (**j**), (**k**), (**l**), (**r**). *$*P ≤ 0.5$, $**P ≤ 0.01$, $***P ≤ 0.001$, $****P ≤ 0.0001$, ns: no significance. Data are the mean ± SEM. Source data are provided as a Source Data file.

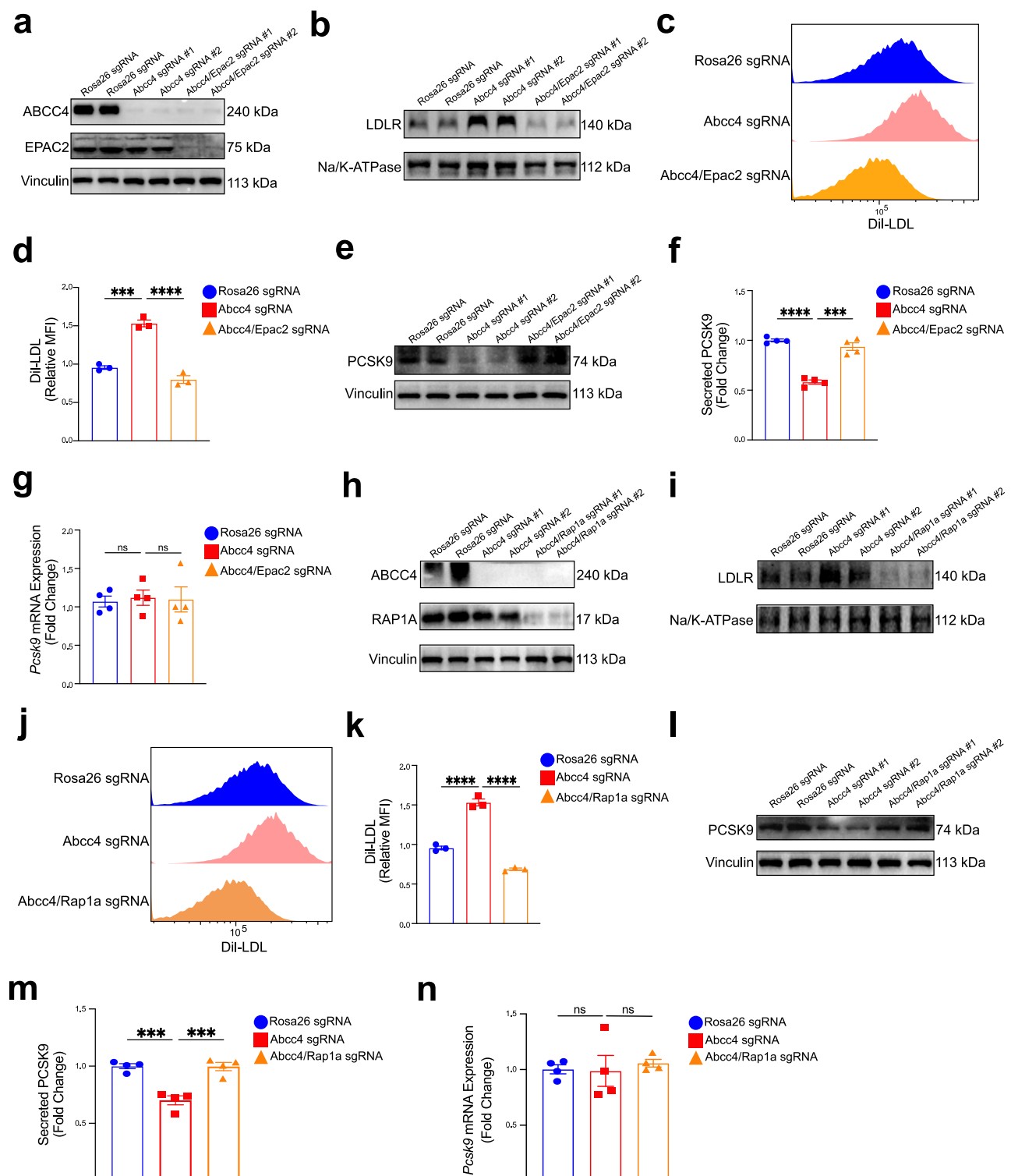

factors (GEFs)[72,73]. Emerging evidence suggests that EPAC proteins, particularly EPAC2, bind to cAMP with high affinity and activate small GTPases of the RAS superfamily, including RAP1 and RAP2[74], consistent with the signaling pathway enriched in our transcriptome data (Fig. 5c-f). To determine whether elevated intracellular cAMP levels mediate ABCC4-dependent regulation via the downstream Epac2/Rap1 signaling, we generated *Abcc4/Epac2* double-knockout (DKO) AML12 cells using two different sgRNAs targeting the gene (Fig. 6a). Interestingly, elevated LDLR surface abundance induced by *Abcc4* deficiency was abrogated in *Abcc4/*

*Epac2* DKO AML12 cells (Fig. 6b). Furthermore, when incubated with Dil-LDL particles, *Abcc4/Epac2* DKO cells exhibited impaired LDL uptake, in contrast to the enhanced uptake seen in *Abcc4*-deficient cells (Fig. 6c, d). Importantly, we also observed that lacking both *Epac2* and *Abcc4* in hepatocytes led to an increase in intracellular and secreted PCSK9 protein levels, without significant changes in *Pcsk9* mRNA expression (Fig. 6e–g).

EPAC2 protein not only serves as a key intracellular effector of cAMP but also acts as an upstream activator of the small GTPase RAP1, as supported by our RNA sequencing data (Fig. 5). To further elucidate the role of

**Fig. 6 | ABCC4-cAMP signaling regulating cell surface LDLR availability is dependent on essential roles of Epac2/Rap1a. a** Generation of *Abcc4/Epac2* double-knockout (DKO) AML12 cells using two sgRNAs targeting *Epac2* gene. Immunoblotting analysis of ABCC4, EPAC2 and Vinculin protein expression in AML12 cells (Rosa26 sgRNA, Abcc4 sgRNA #1, Abcc4 sgRNA #2, Abcc4/ Epac2 sgRNA #1, Abcc4/Epac2 gRNA #2). **b** Immunoblotting analysis of LDLR protein expression from the plasma membrane fractions in AML12 cells (Rosa26 sgRNA, Abcc4 sgRNA #1, Abcc4 sgRNA #2, Abcc4/Epac2 sgRNA #1, Abcc4/Epac2 gRNA #2). **c** Flow cytometry analysis of Dil-LDL uptake assay (1 h) in AML12 cells (Rosa26 sgRNA, Abcc4 sgRNA, Abcc4/Epac2 sgRNA). **d** The relative MFI of Dil-LDL quantification were from 3 independent experiments. **e** Immunoblotting analysis of PCSK9 protein expression in AML12 cells (Rosa26 sgRNA, Abcc4 sgRNA #1, Abcc4 sgRNA #2, Abcc4/Epac2 sgRNA #1, Abcc4/Epac2 gRNA #2). **f** Secreted PCSK9 levels in AML12 cells (Rosa26 sgRNA, Abcc4 sgRNA, Abcc4/Epac2 sgRNA) from four independent experiments. **g** Relative *Pcsk9* mRNA expression level in AML12 cells (Rosa26 sgRNA, Abcc4 sgRNA, Abcc4/Epac2 sgRNA) from four independent experiments. **h** Generation of *Abcc4/ Rap1a* DKO AML12 cells using two independent sgRNAs targeting *Rap1a* gene.

Immunoblotting analysis of ABCC4, RAP1A and Vinculin protein expression in AML12 cells (Rosa26 sgRNA, Abcc4 sgRNA #1, Abcc4 sgRNA #2, Abcc4/Rap1a sgRNA #1, Abcc4/Rap1a sgRNA #2). **i** Immunoblotting analysis of LDLR protein expression from the plasma membrane fractions in AML12 cells (Rosa26 sgRNA, Abcc4 sgRNA #1, Abcc4 sgRNA #2, Abcc4/Rap1a sgRNA #1, Abcc4/Rap1a sgRNA #2). **j** Flow cytometry analysis of Dil-LDL uptake assay (1 h) in AML12 cells (Rosa26 sgRNA, Abcc4 sgRNA, Abcc4/Rap1a sgRNA). **k** The relative MFI of Dil-LDL quantification were from 3 independent experiments. **l** Immunoblotting analysis of PCSK9 protein expression in AML12 cells (Rosa26 sgRNA, Abcc4 sgRNA #1, Abcc4 sgRNA #2, Abcc4/Rap1a sgRNA #1, Abcc4/Rap1a sgRNA #2). **m** Secreted PCSK9 levels in AML12 cells (Rosa26 sgRNA, Abcc4 sgRNA, Abcc4/Rap1a sgRNA) from four independent experiments. **n** Relative *Pcsk9* mRNA expression level in AML12 cells (Rosa26 sgRNA, Abcc4 sgRNA, Abcc4/Rap1a sgRNA) from four independent experiments. Statistical analysis was performed by a Welch ANOVA test followed by a post hoc analysis using the Tamhane T2 method in (**d**), (**k**); a ordinary one-way ANOVA followed by Bonferroni's multiple comparison test in (**f**), (**g**), (**m**), (**n**). *$P \le 0.5$, **$P \le 0.01$, ***$P \le 0.001$, ****$P \le 0.0001$, ns: no significance. Data are the mean ± SEM. Source data are provided as a Source Data file.

RAP1 in regulating LDLR abundance, we generated *Abcc4/Rap1a* double-knockout (DKO) AML12 cells. Double-knockout of *Abcc4/Rap1a* in AML12 cells depleted LDLR protein expression levels on cell membrane, as evidenced by immunoblotting data (Fig. 6h, i). In line with the observed decline in LDLR surface abundance, *Abcc4/Rap1a* DKO cells impeded LDL uptake by LDLR (Fig. 6j, k). Furthermore, *Abcc4/Rap1a* DKO hepatocytes displayed a marked increase in both intracellular and secreted PCSK9 protein levels, while *Pcsk9* mRNA expression remained unchanged (Fig. 6l–n). Overall, these data highlight the pivotal role of the ABCC4-cAMP-Epac2/Rap1a signaling pathway in modulating surface LDLR availability and LDL-C uptake in hepatocytes.

## Discussion

In this study, we performed FACS-based genome-scale CRISPR screens to identify critical modulators of hepatic surface LDLR abundance. Two independent biological replicates of this screen demonstrated strong concordance, consistently highlighting top hits that enhanced LDLR surface levels in hepatocytes. Among these, ABCC4 emerged as the leading candidate, with its sgRNAs highly enriched in the LDLR[high] subpopulation. Functional validation revealed that ABCC4 knockout markedly increased surface LDLR abundance in both *murine* and *human* hepatocytes, independent of LDLR transcription. Additionally, we showed that hepatic-specific disruption of *Abcc4* in mice similarly elevated membrane LDLR protein levels in the liver and reduced plasma LDL-C levels. Pharmacological inhibition of ABCC4 using the specific inhibitor Ceefourin-1 replicated these effects under both NCD and HFD conditions in vivo. Intriguingly, we observed that Ceefourin-1 treatment greatly reduced liver TC levels, improved hepatic insulin sensitivity, and ameliorated glucose homeostasis dysregulation in HFD-fed mice. Mechanistically, increased LDLR surface availability was driven by altered intracellular cAMP distribution and the activation of its downstream Epac2/Rap1a signaling, upon *Abcc4* deficiency. Activation of cAMP signaling pathway due to *Abcc4* deficiency suppressed hepatic PCSK9 protein levels, thereby preventing LDLR lysosomal degradation and maintaining systematic LDL-C homeostasis.

Pooled CRISPR screening is a practical approach for genome-scale, comprehensive analysis of selectable phenotypes using florescence-activated cell sorting (FACS) in the fields of metabolic and cardiovascular medicine[75–77]. Researchers have previously conceived the project by designing a lentiviral sgRNA library targeting the human LDLR gene to identify potential candidate variants[78]. Moreover, experimental evidence has recapitulated key modulators of cellular LDL uptake, as well as post-transcriptional and posttranslational regulators of LDLR[79–82]. Hence, we conducted a genome-scale screening in hepatocytes using a library targeting over 18,000 protein-coding genes to identify genes whose deficiency culminated in the LDLR[high] phenotype. The top 10% of LDLR[high]-expressing cells were isolated via anti-LDLR antibody staining and FACS. MAGeCK

analysis of enriched sgRNAs identified *ABCC4* as a top-scoring candidate gene with significant effects on surface LDLR abundance (Fig. 1c). *Abcc4/ ABCC4* knockout exerted a regulatory effect on surface LDLR abundance in hepatocytes, thereby promoting LDL uptake. In addition to novel modifiers, *LDLR*[53,55], *SREBF2*[80] and *MBTPS1*[83] (membrane-bound transcription factor peptidase, site 1) were identified as positive hits in our screening data, though they did not exhibit high ranking or statistical significance.

ABCC4 (MRP4) is an ATP-dependent efflux transporter of the ABC family that exports a broad range of substrates, including anticancer drugs, bile acids, and cyclic nucleotides[43,50,84–86]. Much attention has been focused on the association between *ABCC4* gene and leukemia outcomes, as well as platelet activation[87,88]. ABCC4 is highly expressed in the liver and is localized to the basolateral hepatocyte membrane, where it mediates the transport of molecules into the bloodstream[51,89,90]. In our study, genetic deletion of *Abcc4/ABCC4* enhanced cell surface LDLR protein expression and LDL uptake without affecting sterol sensing, cholesterol synthesis or *LDLR* mRNA levels. Accompanied by two individual approaches to silence ABCC4 function in vivo, we further reinforced the idea that ABCC4 inhibition increased hepatic membrane LDLR expression and lowered plasma LDL-C levels. These findings highlight ABCC4 as a key regulator of LDLR protein expression and cholesterol metabolism, and a promising therapeutic target for hyperlipidemia and the reduction of cardiovascular risk. However, we observed that ABCC4 inhibition also significantly reduced liver TC levels and improved hepatic insulin resistance in HFD-fed mice. While the mechanisms underlying these metabolic benefits remain unclear, one possibility is that loss of ABCC4 function may modulate glucagon signaling, thereby contributing to a coordinated regulatory effect on metabolic abnormalities and related diseases. Future studies are warranted to elucidate the role of ABCC4 in metabolic disorders such as insulin resistance, obesity, diabetes and hyperlipidemia.

Regarding to the precise mechanistic explanation for ABCC4, our transcriptomic analysis and cAMP measurements revealed that *Abcc4* deficiency altered cAMP distribution, thereby activating the intracellular cAMP signaling pathway. Consistent with earlier studies showing that cAMP signaling regulates key physiological processes, including gonadotropin-releasing hormone secretion, insulin granule dynamics and cardiac function[91–94], our data pointed toward EPAC proteins as key effectors. EPAC1 and EPAC2 contain both a cAMP-binding domain and a guanine nucleotide exchange factors (GEFs) domain, enabling activation of RAS superfamily small GTPases Rap1[72–74,95]. Several lines of evidence has conceived that EPAC2 plays significant roles in the liver[45,71,96], endocrine glands and brain[97–99]. In line with this, our sequencing data illustrated that *Abcc4* deficiency or inhibition in hepatocytes activated a broad spectrum of genes involved in cAMP and Rap1 signaling pathways (Fig. 5d–f; Fig. S6). In this context, we propose that altered intracellular cAMP distribution in *Abcc4*-deficient hepatocytes suppressed PCSK9 protein levels via its

downstream Epac2/Rap1a signals. To gain deeper exploration of ABCC4-cAMP-Epac2/Rap1a signaling in regulating LDLR surface abundance, double-knockout of *Abcc4/Epac2* and *Abcc4/Rap1a* in AML12 cells led to a decrease in membrane LDLR protein levels and an increase in PCSK9 protein levels. Collectively, our results highlight a critical role of ABCC4 in lipid metabolism through cAMP-Epac2/Rap1a signaling, which restrains hepatic PCSK9 protein levels, thereby facilitating LDLR surface availability and promoting clearance of circulating LDL-C.

All the data presented in this study support the conclusion that ABCC4 suppresses LDLR availability and accordingly inhibits the clearance of plasma LDL cholesterol. However, several limitations warrant future investigation. First, although our screening identified novel modifiers of surface LDLR protein expression, it failed to prioritize the effects of some canonical regulators. This may reflect a suboptimal signal-to-noise ratio inherent to pooled CRISPR screens, where technical noise, stochastic dropout, or variable knockout efficiency can obscure genes with modest effects. It is possible that sgRNAs targeting PCSK9 or MYLIP may have been underrepresented or less effective in our system, whereas ABCC4 knockout produced a more robust, multi-level effect, yielding a stronger phenotype. Moreover, our screening focused primarily on genes driving substantial transcriptional changes under homeostatic conditions, whereas post-transcriptional regulation also plays an important role in determining protein levels. Future work should directly compare transcriptional versus post-transcriptional regulatory mechanisms, and systematically evaluate the performance variability of individual sgRNAs. Second, while we unexpectedly observed that ABCC4 inhibition significantly reduced liver TC levels and improved hepatic insulin resistance in HFD-fed mice, it is essential to further validate this effect in hepatocyte-specific *Abcc4* knockout mice under metabolic stress. Such studies would clarify the role of ABCC4 in glucose and lipid metabolism, as well as its potential impact on atherosclerotic lesions. Third, although our work proposes ABCC4 inhibitor as a potential therapeutic target for hyperlipidemia, its synergistic effect with statins or PCSK9 inhibitors remains untested. More studies are needed to explore and confirm additive or synergistic effects, which could enhance therapeutic efficacy.

In conclusion, this study provides novel insights into the role of ABCC4 in regulating LDLR surface availability in hepatocytes dependent on cAMP-Epac2/Rap1a signaling, which contributes to the clearance for circulating LDL-C. These findings establish ABCC4 as a promising therapeutic target for hypercholesterolemia and atherosclerotic cardiovascular diseases.

## Methods
### Cell lines and culture conditions
Murine hepatocyte cell line AML12 (Alpha Mouse Liver 12, ATCC CRL-2254) and human embryonic kidney (HEK) 293 T cell line (ATCC CRL-3216) were originally obtained from American Tissue Culture Collection Biobank (Manassas, VA, USA) and were incubated at 37 °C with 5% CO2. Human hepatocyte cell line LO2 were originally purchased from QuiCell Biotechnology (QuiCell-L090, Shanghai, China). AML12 cells, LO2 cells and 293 T cells were cultured in Dulbecco's modified Eagle's medium (DMEM) (Gibco, 11995065) supplemented with 10% heat-inactivated FBS (ExCell, FSP500), penicillin/streptomycin 100 U/ml (Gibco, 15140122) and 2 mM L-glutamine (Gibco, 25030081).

### Genome-wide CRISPR/Cas9 knockout sgRNA library, lentiviral production and transfection
The genome-wide CRISPR library (Liu Mouse CRISPR Knockout Library in lentiCRISPRv2-blast, Addgene#1000000173) was generated by Xiaole Shirley Liu[28]. Libraries were packaged to produce lentiviruses using HEK 293 T cells as described previously[28]. Briefly, HEK293T cells with 90% confluency were co-transfected using 50 μl X-TremeGene Transfection Reagent (Roche, 41106502) diluted in 2.5 ml of Opti-MEM (ThermoFisher Scientific, 11058021) that was combined with 10 μg of the plasmid pool DNA and a mixture of 7.5 μg psPAX2 (Addgene#12260) and 3 μg pMD2.G (Addgene#12259). After 12 hours, the medium was replaced, and 72 hours

post-transfection, virus-containing supernatant was collected and filtered through a 0.45-μm filter. The virus was aliquoted and stored at −80 °C until needed. Functional viral titer was determined using hepatocyte cell lines AML12 cells and LO2 cells by measuring survival rates post-transduction with different viral dilutions under blasticidin selection. The required lentivirus quantity, corresponding to a multiplicity of infection (MOI) of 0.3, was calculated for subsequent genome-wide screening transductions.

### FACS-based CRISPR screening approach in murine hepatocyte cell line AML12 cells
Wild-type (WT) murine hepatocyte cell line AML12 were independently screened. For an unbiased approach, approximately 100 million cells were transduced with the Mouse CRISPR Knockout Library viruses at an MOI of ~0.3. For lentiviral transduction, cells were spinfected at 260 rpm for 60 min at 37 °C with 8 μg/mL polybrene (Beyotime, C0351) in medium. Blasticidin S (Beyotime, ST018) was added at 2 μg/mL two days after transduction to eliminate uninfected cells, continuing until selection was complete. Cells were trypsinized, and a partial sample was collected for evaluating the sgRNA coverage. Remaining cells were cultured to maintain the growth phase for the focused screens, ensuring a sufficient cell count. Fourteen days post-transduction, 20 million edited cells were harvested, stained for surface LDLR quantification and sorted using fluorescence-activated cell sorting (FACS) with a BD FACS Melody system. Flow cytometry gating for LDLR expression was defined relative to IgG Isotype control or LDLR-targeted wide-type cells, using Mouse LDLR PE-conjugated Antibody (R&D Systems, FAB2255P). For the focused screens, the top 10% LDLR-positive AML12 cells (LDLR^high subpopulation) were collected. Focused sorted samples were stored at -80 °C for DNA extraction.

### Preparation of genomic DNA for next-generation sequencing and analysis of pooled CRISPR screening
Genomic DNA of LDLR^high cells and unsorted control cells were extracted respectively using cell lysis buffer (400 mM Sodium chloride, 10 mM Tris, 2 mM EDTA, 0.5% SDS, pH 8) and then subjected to the established procedures[100]. After barcoding of PCR amplicons, sgRNA sequences were amplified and sequenced using the Illumina HiSeq 2500 platform. The sequencing was performed by Azenta Life Sciences Company (Suzhou, China). Data quality, read counts and identification of significant hits in this screen were analyzed using the Model-based Analysis of Genome-wide CRISPR-Cas9 Knockout (MAGeCK) algorithm[29,30].

### Knockout of candidate genes by CRISPR Cas9 genome editing
To verify our screening results, two independent guide sequences were designed targeting *Abcc4* gene. The sequence of the two *Abcc4* sgRNA are: 5'-ATGCTGCCGGTGCACACCG-3', 5'-CCATGGGGAAGACAACCAC-3'. Each pair of oligonucleotides were annealed and ligated into the lentiCRISPR v2 vector (Addgene#52961). Lentiviruses were produced and used to infect AML12 cells as mentioned above. Following infection with lentivirus for 48 h, the cells were selected by at a concentration of 2 μg/mL puromycin. Knockout editing of gene was confirmed by immunoblotting experiments. ABCC4 genome editing was performed similarly in the human hepatocyte cell line LO2. The sequence of the two *ABCC4* sgRNA are as follows : 5'-CCATGGGGAAGACAACCAC-3', 5'-GGCTGTGATCACACTGCCG-3'.

### Generation of double knockout of *Abcc4/Epac2* and *Abcc4/Rap1a* in murine hepatocyte cell line AML12 cells
To investigate the downstream signals affected by altered intracellular cAMP level, we relatively generated double knockout of *Abcc4/Epac2* and *Abcc4/Rap1a* murine hepatocyte cell line AML12 by CRISPR Cas9 genome editing. The CRISPR Cas9 genome editing in the two genes were performed as above mentioned. The single guide RNA sequences of two genes were listed in the Supplementary Table 3. Each pair of oligonucleotides were annealed and ligated into the lentiCRISPR v2-Blast vector (Addgene#83480). Lentiviruses were produced and used to infect Abcc4-

knockout AML12 cells and the infected cells were selected by Blasticidin S at a concentration of 2 μg/ml. Knockout editing of gene was eventually confirmed by immunoblotting experiments.

## Animal experiments

All animal experimental procedures were approved by the Ethics Committee of Shanghai Sixth People's Hospital Affiliated to Shanghai Jiao Tong University School of Medicine. We have complied with all relevant ethical regulations for animal use. Six-week-old male wide type (WT) C57BL/6 mice were purchased from Shanghai Laboratory Animal Center (SLAC). Mice were fed with either a normal-chow diet (NCD) or 60% high-fat diet (HFD) for 4 weeks, 60% HFD (5.24 kcal/g with 20% energy derived from protein, 60% from fat, and 20% from carbohydrate; D12492; Research Diets, Inc., New Brunswick, New Jersey). A rodent diet containing 60 kcal% fat is typically used for studies related to obesity, metabolic disorders, and lipid-related disorders, as it closely mimics high-cholesterol diets found in human populations[101–103]. Animals were housed at approximately 21°C with 55% humidity under a 12-hour light/12-hour dark cycle. Ceefourin-1 (Medchem Expression, HY-101453) or vehicle (DMSO and corn oil) was administered intraperitoneally three times weekly at 10 mg/kg body weight for 4 weeks. All mice were euthanized (cervical dislocation) at the end of the experiment, and liver tissues were extracted for further investigation.

## Construction of adeno-associated virus

Adeno-associated virus 8 serotype (AAV8), packaged by Genechem (Shanghai, China), were used to manipulate ABCC4 expression in mice livers. We chose recombinant AAV serotype-8 expressing the Abcc4 RNAi under the liver-specific apolipoprotein E/human alpha 1-antitrypsin promter [AAV8-ApoE/hAAT; $2 \times 10^{11}$ viral genomes (VG)] for suppressing the expression of ABCC4 in hepatocytes, which were administered through tail vein injection. The ApoE/hAAT promoter is known for its strong and liver-specific expression, which has been demonstrated in previous studies to provide effective gene regulation specifically within hepatocytes[56,57,104]. We and technicians believe that this promoter is particularly well-suited for models where a hepatocyte-specific expression pattern is required for investigating liver-specific gene functions, especially when combined with RNA interference strategies. The *Abcc4* gene sequence (NM_001033336) was targeted using synthesized small pieces of interfering double-stranded RNA (RNAi) construct containing the sequence: GAGTTTCTGAAATCTGGTGTA (AAV_Abcc4_RNAi). A control RNAi construct targeting green fluorescent protein (GFP; AAV_GFP_RNAi) contained the sequence: TTCTCCGAACGTGTCACGT. Six-week-old Wild Type (WT) C57BL/6 male mice were administered either AAV_Abcc4_RNAi (AAV-ApoE/hAAT-Abcc4_RNAi) or negative control AAV_GFP_RNAi (AAV-ApoE/hAAT-GFP_RNAi) through tail vein injection.

## Intraperitoneal glucose tolerance test (IPGTT) and insulin tolerance test (IPITT)

At the end of the 4th week of the animal experiment, mice were subjected to glucose and insulin tolerance tests. For the intraperitoneal glucose tolerance test (IPGTT), mice were fasted for 6 hours, and baseline blood glucose levels were measured via tail vein using a glucometer. Glucose was then administered intraperitoneally at a dose of 2 g/kg body weight, and blood glucose levels were recorded at 0, 15, 30, 60, 90, and 120 minutes post-injection. For the insulin tolerance test (ITT), mice were fasted for 4 hours and subsequently injected intraperitoneally with insulin (0.75 IU/kg). Blood glucose measurements were taken at 0, 15, 30, 60, 90, and 120 minutes following insulin administration to assess insulin sensitivity.

## Sample preparation and biochemical analysis

Blood and tissue samples were collected immediately following the completion of the animal experiments for biochemical analysis. Serum was isolated by centrifugation of blood samples at 12,000× g for 10 min at 4°C. Liver tissue (approximately 100 mg) was homogenized and then centrifugated at 2500× g for 10 minutes at 4°C to collect the supernatant. The concentrations of total cholesterol (TC), low-density lipoprotein cholesterol (LDL-C) and triglycerides (TG) in serum and liver TC, TG levels were measured using commercial kits (Nanjing Jiancheng Bioengineering Institute, Nanjing, China). Additionally, serum levels of PCSK9 (Abcam, ab215538), as well as serum insulin (Nanjing Jiancheng Bioengineering Institute, Nanjing, China) were quantified using commercially-available detection kits, according to the manufacturers' protocols. Lipids were extracted from the AML12 cells by adding 0.2 mL of phosphate buffer for homogenization, followed by ultrasonic disruption in an ice water bath. The resulting homogenate was directly used to measure cellular cholesterol contents using commercial kits (Nanjing Jiancheng Bioengineering Institute, Nanjing, China). The PCSK9 content in the cell supernatant was measured using the same commercial kit (Abcam, ab215538).

## Histological analysis

Hematoxylin and eosin (H&E) and Oil Red O staining was performed on formalin-fixed or frozen liver sections. LIver tissues were fixed in 4% paraformaldehyde (Servicebio, Wuhan, China, G1101), embedded in paraffin, and sectioned. Then Hematoxylin and eosin (H&E) staining was performed to observe tissue morphology and pathological changes. Another part of liver tissues were embedded in OCT, frozen, sectioned, and stained with Oil Red O to assess lipid deposition.

## Western blotting

Whole lysates and membrane fractions from cultured AML12 cells, LO2 cells and liver tissue were extracted by using RIPA lysis Buffer (Beyotime, P0013B) and Membrane and Cytosol Protein Extraction Kit (Beyotime, P0033) respectively. For the western blotting experiment, equivalent protein samples were separated by 7.5% or 10% SDS-PAGE and transferred onto polyvinylidene fluoride (PVDF) membranes. Membranes were blocked with 5% non-fat milk at room temperature for 1 hour, followed by incubation with primary antibody overnight at 4°C. After 3 washed with TBST, membranes were incubated with secondary antibody at room temperature for 1 hour and visualized using an electrochemiluminescence (ECL) system. The primary antibody used were as follows: anti-LDLR at 1:1000 (Abmart, T55235); ABCC4 at 1:1000 (Abcam, ab15602); PCSK9 at 1:1000 (Proteintech, 27882-1-AP); EPAC2 at 1:1000 (Proteintech, 19103-1-AP); RAP1A at 1:1000 (Proteintech, 16336-1-AP); Vinculin at 1:5000 (Sigma, V9131); Na/K-ATPase at 1:5000 (Abmart, T55159).

## RNA isolation and real-time qPCR

Total RNA was extracted from mouse livers or cultured AML12 cells and LO2 cells using TRIzol reagent (Sigma-Aldrich, T9424) following standard protocols. RNA concentrations and purity were assessed by UV/VIS Nano Spectrophotometer (Cobra, Oxford). mRNA was reverse transcribed to cDNA using the HiScript II Reverse Transcriptase Reagent kit with gDNA eraser (Vazyme, China, R222-01). Quantitative RT-PCR were performed with AceQ Universal SYBR qPCR Master Mix (Vazyme, China, Q511-03) on a LightCycler® 96 Instrument (Roche, USA). Primers sequences are provided in Supplementary Table 4.

## RNA-sequencing and bioinformatics analysis

AML12 cells were used for RNA isolation, library construction, and RNA sequencing. Total RNA was extracted using TRIzol reagent (Sigma-Aldrich, T9424) following the manufacturer's protocol. RNA quality and integrity was assessed with a Bioanalyzer 2100 (Agilent, CA, USA; RIN > 7.0) and confirmed by denaturing agarose gel electrophoresis. cDNA libraries were constructed and sequenced on an illumina Novaseq™ 6000 (LC-Bio Technology CO., Ltd., Hangzhou, China). Pair-end FASTQ files were mapped to the Mus_musculus.GRCm38.101 reference genome using the HISAT2 package. FPKM values were calculated with StringTie and Ballgown (http://www.bioconductor.org/packages/release/bioc/html/ballgown.html). Differential gene expression was analyzed using DESeq2, with genes having a $P$ value ≤ 0.05 and absolute fold change ≥2 recognized as differentially

expressed (DEGs). Heatmaps were generated, and Gene Ontology (GO) and KEGG pathway analyses were performed on DEGs. Gene set enrichment analysis (GSEA) was conducted using GSEA (v4.1.0) and MSigDB to identify gene sets involved in specific pathways.

## Dil-LDL uptake

Cultured AML12 cells were precooled to 4℃ for 30 minutes and incubated with 20ug/ml Dil-LDL (3,3'-dioctadecylindocarbocyanine LDL; Yiyuan biotechnology, YB-0011) at 4℃ for 1 hour to allow binding. The cells were then washed three times with ice-chilled PBS and shifted to 37 ℃ for 3 hours to enable uptake. Afterward, cells were again washed, trypsinized into single cell suspension, and analyzed by flow cytometry with appropriate excitation and emission wavelengths following the suggested protocol. Immunofluorescence microscopy imaging was performed as above. Cells were plated on 35 mm Confocal Dishes, imaged using a 63× lens under oil immersion, and captured with a Nikon A1 Confocal Microscope System.

## ELISA cAMP assays

Intracellular and extracellular cAMP levels were measured in cultured lysates and medium of Abcc4_KO and Control_KO AML12 cells based on the competition between HRP-labeled cAMP and free cAMP according to colorimetric ELISA cAMP Assay Kit (Abcam, ab234585). Altered cAMP distribution in AML12 cells treated with DMSO or Ceefourin-1 was also evaluated as above.

## Statistical and reproducibility

Continuous data are presented as means ± SEM. Statistical analysis was conducted using Prism 10 (GraphPad Software). For comparison between two groups, an unpaired two-tailed Student's t-test was used, assuming equal variance. If variances were unequal, t test with Welch's correction was applied. Similarly, when data were compared among more than 3 groups, a ordinary one-way ANOVA was followed by Bonferroni's multiple comparisons test based on the equal variance. Providing that there were significantly equal variance among more than 3 groups, Brown-Forsythe and a Welch ANOVA test was performed, using the Tamhane T2 method. Statistical significance was defined ad $P$ values < 0.05. Representative images displayed in figures were selected from at least three biologically independent experiments.

## Reporting summary

Further information on research design is available in the Nature Portfolio Reporting Summary linked to this article.

## Data availability

All the data supporting the findings in this study are included in the main article and its supplementary data files. The processed sequencing data in this paper have been deposited in NCBI GEO database (GSE297526). Source data behind all graphs are provided in Data S6. Uncropped, unedited blot images and the gating strategy of flow cytometry (FACS) plots in this paper are provided in Figs. S9 and S10, respectively. Further information and requests for reagents may be directed to, and will be fulfilled by Prof. Xiaoqing Wang (Xiaoqing_Wang@uestc.edu.cn).

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

## Acknowledgements

We thank Congfeng Xu for assistance with fluorescence-activated cell sorting analysis. This study was supported by the National Natural Science Foundation of China (No. 8217051361 and 82570533) and the Shanghai Sixth People's Hospital Institute-level project. Additionally, we would like to thank LC-BIO TECHNOLOGIES (HANGZHOU) Co., Ltd and Mr. Run Qin for providing transcriptome sequencing technologies.

## Author contributions

Xiaoqing Wang developed the concept and designed this work. Jiaxin Chen, Hui Huang and Chi Chen performed the experiments and wrote the manuscript draft. Guofang Xia provided assistance for fluorescence-activated cell sorting analysis. Hao Huang, Peng Luo, Yu Chen and Jinsong Li carried out the data acquisition. Lu Li, Yan Xiong, Jing Lin, Chenzhang Ji, Guangre Xu and Liang Wen performed data analysis. Jin Zhou contributed to the advice. Wenjie Tian, Peng Wei, Chengxing Shen and Xiaoqing Wang edited and revised the manuscript. All authors read and approved this manuscript.

## Competing interests

The authors declare no competing interests.
