## [Transparent Peer Review file · Communications Biology]

ABCC4 impairs the clearance of plasma LDL cholesterol through suppressing LDLR expression in the liver

Corresponding Author: Professor Xiaoqing Wang

This manuscript has been previously submitted at another journal. This document only contains information relating to versions considered at Communications Biology.

Version 0:

Reviewer comments:

Reviewer #1

(Remarks to the Author)

In this study, Chen et al. identified Atp-binding cassette c4 (Abcc4) transporter as a new actor in controlling hepatic surface expression of the low-density lipoprotein (LDL) receptor through a mechanism involving the regulation of proprotein convertase subtilisin/kexin type 9 (PCSK9). Such a mechanism could be of major interest since a low expression of LDL-R is associated with hypercholesterolemia, a strong risk factor for atherosclerotic cardiovascular diseases (ASCVD). This study is very interesting since targeting PCSK9 is now a major therapeutic strategy for reducing circulating cholesterol concentration in patients with familial hypercholesterolemia. In this context the deciphering of a new pathway controlling the PCSK9 expression may open a new therapeutic strategy in familial hypercholesterolemia. However, several major points reduce the enthusiasm and interrogate on the specificity of this new pathway.

Major points.

Introduction. This part is too long and must be shortened. More importantly the objective of the study is weakly defined in this section (line 78) but is more convincing at the beginning of the "Result" section (line 96). The authors should reinforce this point in the "Introduction" section and prevent redundancy with the "Result" section.

What is the pattern of expression of Abcc4 in tissues and cells in both mice and humans ? Is Abcc4 constitutively expressed in all cells and tissues ? What is known on the regulation of Abcc4 expression ? Is the Abcc4 gene regulated by cholesterol or lipids ? The authors should give more information on this transporter.

Is there any association between ABCC4 SNPs and cholesterol levels and/or ASCVD in human ? More importantly, what is the localization of Abcc4 expression in hepatocytes ?

Figure 1. Although the MAGeCK strategy to identify potential targets controlling LDLR expression is interesting, the Reviewer has some doubts on its efficiency since major genes known to control LDLR expression such as PCSK9 and MYLIP seem to not be statistically identified (Supplemental Tables). However, the authors indicated in the Discussion section (line 308) that "In addition, a subset of negative hits such as PCSK9 and MYLIP were exactly confirmed by using our individual sgRNAs". This point is not clear and must be detailed by the authors and data must be shown.

Figure 5. In addition, invalidation of Abcc4 in AML12 cells was accompanied by a modification of several biological pathways by GSEA with none on cholesterol metabolism suggesting that Abcc4 did not impact cholesterol uptake and metabolism in hepatocytes. The authors must discuss this discrepancy with their proposed mechanism.

Figure 2. Although the impact of Abcc4 on the cell surface expression of LDLR and the LDL uptake is convincing, the authors however failed to prove that those effects led to an increased cholesterol content into the cell. Then, intracellular content should be quantified and the expression of cholesterol-dependent genes including SREBP2 and HMGCoAR should be analyzed.

Figure 3-4. In vivo experiments in mice conducted in this study are intriguing. First of all, it is very surprising that a high-fat

diet and not a high-cholesterol diet was used in these experiments to validate the role of Abcc4 on cholesterol uptake by the liver. What is the composition of the HFD used ? In addition, why the authors did not use a more appropriate model for hypercholesterolemia such as ApoE KO mice (the use of Ldlr KO mice being not possible here)? This mouse model would have been very useful for studying the impact of Abcc4 depletion on atherosclerosis development. Finally, the authors must explain why they selected the ApoE/hAAT promoter for the expression of RNAi targeting Abcc4 “specifically in hepatocyte”? The use of largely used promoters such as Albumin or TBG would have been more appropriate. The authors must discuss this point. Was the same promoter used for the expression of control GFP_RNAis ?

Figure 3. Circulating cholesterol is mostly carried by HDL and not LDL in the mouse model and the diet used in this study. This very likely explains why the authors observed no significant changes in plasma total cholesterol levels between AAV8_GFP_RNAi controls and AAV8_Abcc4_RNAi mice. Isolation of lipoproteins would help to determine if cholesterol associated to LDL (LDL-C) was reduced as expected upon hepatic Abcc4 deficiency. Surprisingly LDL-C quantification (although using a questionable assay) was performed in mice treated with Ceefourin-1 (Figure 4g) and the Reviewer wonders why this was not done in Abcc4 deficient mice. A similar comment can be made with Oil Red O (ORO) staining in the liver (Figure 4h). However, a direct quantification of cholesterol in the liver would be more appropriate since ORO staining reflects both cholesterol and triglyceride accumulation in the liver.

Figure 4. As a follow up of the previous comment, how do the authors explain the reduction of neutral lipid accumulation in the liver from mice upon treatment with the Abcc4 inhibitor, Ceefourin-1 ? Based on the mechanism proposed by the authors for Abcc4, an increased ORO staining would be expected (increased LDLR-dependent uptake of cholesterol). These data questioned on the specificity of Ceefourin-1. The authors must clarify this point.

Figure 5C. Transcriptome analysis strongly suggests that biological pathways linked to diabetes and insulin resistance rather than cholesterol metabolism are modulated upon Abcc4 deficiency. This could explain in part the reduced lipid accumulation in the liver upon HFD diet in mice treated with Ceefourin-1. Were insulin and glucose tolerance tests performed in mice ? What about fasting plasma insulin and glucose levels ? Plasma triglyceride levels ? This point is critical.

Figures 5i-k. Was secreted PCSK9 levels also reduced in AML12 cells treated with Abcc4 sgRNAs ?

Figure 2j and 6d. It would be very elegant to show that addition of recombinant PCSK9 abolishes the increased uptake of DiI-LDL in Abcc4 deficient cells. This would reinforce the role of Abcc4 in the LDLR-dependent uptake of cholesterol.

Figure 6b, line 260. “Interestingly, elevated LDLR surface abundance induced by Abcc4 deficiency was abrogated in Epac2 knockout AML12 cells”. This assertion is not true since no experiment was performed with a combined inhibition of both Abcc4 and Epac2 by sgRNA. As a whole, the authors failed to validate the requirement of the Epac2 / Rap1 pathway in the reduction of PCSK9 expression and DiI-LDL uptake upon Abcc4 deficiency. This needs to be validated in experiments with both single and dual silencing of Abcc4 and either Epac2 or Rap1, respectively.

Discussion section, line 360. As mentioned above, this conclusion is not supported by data presented in this manuscript.

Minor points. “stains” must be replaced by “statins” in several lines of the manuscript (lines 76, 357...)

Reviewer #2

(Remarks to the Author)

The manuscript of Chen et al. describes the discovery of the protein ABCC4/MRP4 as a novel modulator of cell surface low density lipoprotein (LDL) receptor (LDLR) in AML12 hepatocytes. The authors provide evidence that intracellular cAMP levels regulate PCSK9 levels, which regulate cell surface LDLR. The authors supported this data with animal work and work with a specific ABCC4 inhibitor ceefourin-1. Overall I find the work of good quality and conclusions are well supported by the results. Spelling and grammar are poor, with lots of mistakes in the text. The biggest problem is that this pathway has already been published and cited (reference 43), though the authors fail to properly present this paper. While I find the work with ceefourin-1 novel, overall this paper lacks novelty. My comments are below.

1. The authors should describe a little about the proteins that were discovered in the initial screen, in addition to MDR4/ABCC4. There is certainly no cohesion in the proteins or pathways identified.
2. Line 174-175: What are the levels of LDL-C in this mouse model? The likely reason that the effect on total cholesterol is not significant is because, LDL-C makes up a small proportion of the total cholesterol.
3. Line 183: Did the authors mean cytosolic? If ABCC4 is a cAMP transporter, I assumed that it was externalizing cAMP.
4. Line 221-222: As a philosophical question, if ABCC4 knockout causes a wide range of alterations in gene expression, then would targeting ABCC4 as a therapeutic treatment be sensible?
5. Line 246-248: The authors have not proven a correlation between cAMP levels and ABCC4. They are coincident. It is possible that ABCC4 has other functions besides cAMP transport, or that ceefourin-1 has other targets or effects. While I agree that the evidence has suggested an effect of ABCC4 on PCSK9, there is no evidence to support the effect of cAMP on PCSK9. In no way has the evidence proven anything, rather the evidence is supportive of a model where...
6. Line 263-275: The authors have provided good evidence to support the role cAMP and Epac in affecting LDLR surface presentation. The authors postulate that PCSK9 plays a role. As PCSK9 mRNA is not altered, but protein levels are affected, what is the hypothesis then? Impaired trafficking of PCSK9 so that it is not secreted? Phosphorylation of PCSK9?
7. A published paper from 2018 has already established that Epac and Rap-1 affect PCSK9 protein levels (<https://www.ahajournals.org/doi/10.1161/CIRCRESAHA.118.313648>). This is reported as reference 43. In this paper, the

authors suggest that glucagon signaling affects PCSK9 levels. I believe that the authors have simply tied into this already established pathway. This paper must be properly described in the text. Furthermore, the authors should perform experiments with glucagon to either prove or disprove that the findings of this paper support previously published work. Furthermore, another paper has already established that MRP4/ABCC4 is tied to obesity and type 2 diabetes (<https://pmc.ncbi.nlm.nih.gov/articles/PMC8628361/>). So, this paper lacks novelty, except with the treatment with ceefourin-1.

8. Does ABCC4 affect IDOL levels? Is LDLR ubiquitinated? IDOL/MYLIP needs to be ruled out.

Reviewer #3

(Remarks to the Author)

In this paper the authors performed genome wide screens in order to identify novel regulators of LDL receptor expression which could potentially be used as targets for LDL lowering therapies. Using this approach the authors focused on the top candidate which is the membrane transporter Abcc4. Using inhibitors and in vitro/in vivo studies they propose that Abcc4 inhibition leads to upregulation of the LDLR gene via a pathway that includes cAMP, the cAMP sensor Epac, the small GTPase Rap1 and PCSK9.

Overall, the experiments are well designed and controlled but the main weakness of the paper is that it lacks sufficient mechanistic confirmation regarding the proposed signal transduction pathway that leads to the non-transcriptional inhibition of PCSK9. For instance no evidence is provided regarding the activation of the small GTPase Rap1 and the same applies to the other effectors of this pathway, including PCSK9. Based on the above, the data in this paper should be considered as preliminary. The authors should also check the manuscript thoroughly for grammar errors and typos.

Version 1:

Reviewer comments:

Reviewer #1

(Remarks to the Author)

The authors have provided a large amount of work in order to address the Reviewer's comments. The new version of the manuscript is more convincing and new data support the mechanism proposed by the authors. Then, most of critical points were clarified but some still need to be elucidated. To note that it was quite difficult to review the revised version of the manuscript without the line numbering. Because of the addition of numerous new data, the manuscript must be thoroughly checked before further editing process (few typos and spelling errors, check the numbering of Figures.....).

Few previous points need further explanations :

Introduction. "Besides PCSK9 antibodies, current clinical therapies including statins and NPC1 (NPC Like Intracellular Cholesterol Transporter 1) inhibitors.....". I guess that NPC1 must be replaced by NPC1L1.

Previous point#3. The new Extended Data Figure 5 is interesting but the legend must be further described. What is the meaning of the adjusted p value in a) ? Are those SNPs associated with plasma lipids (if yes, which ones ? LDL-C ?) or cardiovascular risk (how was defined this risk ?) ? Abbreviations used in b) must be detailed.

Previous point#4. "Specifically, a reason for missing MYLIP or PCSK9 is the optimization of our screening towards the discovery of loss-of-function effects." This point is not clear for me. A loss-of function in PCSK9 is supposed to have similar effects than in ABCC4 (Reduced LDL-C and increased LDL-R in hepatocytes). Then PCSK9 should have been detected at a similar level than ABCC4. Is it correct ?

Previous point#5. Once again, the answer to this question is not clear. What do the authors mean by "undesirable weakness of RNA-seq technology" ? The new data presented in (Extended Data Fig. 2h and 2i) are intriguing and seem not to support a role of ABCC4 in the expression of LDL-R at the cell surface. It is indeed unclear how an increased LDL uptake in AML12 cells is not accompanied by an elevated intracellular cholesterol content. Do the author envisage a compensatory pathway which might mask this effect (cholesterol export, synthesis of biliary acids.....) ? The authors must discuss this potential discrepancy. Indeed, the proposed explanation is poorly convincing "These data suggested that ABCC4 probably only regulates LDL cholesterol through LDLR, not total cholesterol".

Other point, I do not understand the relevance to quantify LDL-C in the intracellular compartment as LDL are secreted by hepatocytes. Finally, the title of the Extended Data Fig. 2h must be corrected ("cholesterol" and not "cholesterol").

Previous point#8. The authors did not answer clearly to this question : "Surprisingly LDL-C quantification (although using a questionable assay) was performed in mice treated with Ceefourin-1 (Figure 4g) and the Reviewer wonders why this was not done in Abcc4 deficient mice. A similar comment can be made with Oil Red O (ORO) staining in the liver (Figure 4h)".

Previous points#9-10. The authors must conclude on the results from these interesting additional experiments. Does the increased ORO staining observed in liver from mice fed a HFD and treated with Ceefourin-1 result from an elevated glucose uptake and lipogenesis that could lead to a higher TG accumulation ? Once again, quantification of triglycerides in both plasma and liver would be helpful to elucidate this point.

Other point : Bar legend from Extended Data Fig. 4e must be corrected (HFD was replaced by NCD in triangles).

Reviewer #2

(Remarks to the Author)

The authors have done a good job of addressing my concerns, with one exception.

There is some confusion about what LDL-C is. LDL-C is the cholesterol contained within LDL. It is a calculated measurement from the Friedewald formula. So, this is a plasma measurement of cholesterol contained within LDL lipoproteins. My confusion comes when the authors discuss "liver LDL-C" in Figure 3h, for example. If LDL is in the plasma, how can we measure liver LDL-C? This needs to be clarified.

Version 2:

Reviewer comments:

Reviewer #1

(Remarks to the Author)

The authors have clarified all the points raised by the Reviewer.

Minor points.

Legend to Extended Data Fig.5a. I am not sure that a simple Student's test was applied for the testing the association between the SNPs and LDL-C levels. This point must be clarified if necessary.

Line 212: "chanegs" must be corrected and replaced by "changed".

Once again, I think that the manuscript must be thoroughly checked before further editing process (few typos and spelling errors, check the numbering of Figures.....).

Reviewer #2

(Remarks to the Author)

My concerns have been addressed.

Dear editor and reviewers,

On behalf of my colleagues and myself, I would like to submit a revision of our manuscript entitled “**ABCC4 impairs the clearance of plasma LDL cholesterol through suppressing LDLR abundance on hepatic surface**” for consideration of publication in *Communications Biology*. We thank the editors and reviewers for their careful and intensive reading and thoughtful comments on our manuscript. We have carefully taken their comments into consideration in preparing our revision, which has resulted in a paper that is clearer, more compelling. The following summarizes how we responded to reviewer comments is our responses to their comments.

Responses to the comments of the reviewers

Reviewer #1 (Remarks to the Author):

In this study, Chen et al. identified Atp-binding cassette c4 (Abcc4) transporter as a new actor in controlling hepatic surface expression of the low-density lipoprotein (LDL) receptor through a mechanism involving the regulation of proprotein convertase subtilisin/kexin type 9 (PCSK9). Such a mechanism could be of major interest since a low expression of LDL-R is associated with hypercholesterolemia, a strong risk factor for atherosclerotic cardiovascular diseases (ASCVD).

This study is very interesting since targeting PCSK9 is now a major therapeutic strategy for reducing circulating cholesterol concentration in patients with familial hypercholesterolemia. In this context the deciphering of a new pathway controlling the PCSK9 expression may open a new therapeutic strategy in familial hypercholesterolemia.

However, several major points reduce the enthusiasm and interrogate on the specificity of this new pathway.

Answer: We thank the reviewer for the excellent comments which we believe can improve the manuscript. And, we have revised the manuscript in close conformity with the requests of the reviewer.

Major points.

1. Introduction. This part is too long and must be shortened. More importantly the objective of the study is weakly defined in this section (line 78) but is more convincing at the beginning of the “Result” section (line 96). The authors should reinforce this point in the “Introduction” section and prevent redundancy with the “Result” section.

Answer: We thank the reviewer for this thoughtful and constructive comment. We have shorten the “Introduction” part and reinforced the objective of our study in this part (line 91-114).

As a transmembrane glycoprotein, low-density lipoprotein receptor (LDLR) is widely expressed on the surface of hepatocytes, where it facilitates the uptake of LDL cholesterol and maintains the homeostasis of lipid metabolism^{20,21}. Approximately, 75% of circulating LDL particles are cleared by LDLR through receptor-mediated endocytosis^{22,23}. As the pH in the endosome decreases, LDLR undergoes characteristic changes of conformation. Then, LDLR recycles back to the hepatocyte surface, whereas LDL-C is transported to lysosomes for

degradation²⁴⁻²⁷. Over the past decade, functional genomic approaches have emerged as powerful tools to systematically identify novel regulators involved in metabolic processes and cardiovascular diseases. Accordingly, we hypothesize that additional novel and more effective modulators of LDLR remain to be identified through CRISPR-Cas9 functional screening.

In this study, we conducted genome-wide screenings by introducing CRISPR libraries containing single guide RNAs (sgRNAs) targeting over 18,000 protein-coding genes into the hepatocyte cell line AML12. Using this approach, we systematically identified key modulators of LDLR protein expression. Notably, we discovered *ABCC4* as a promising hit gene governing LDLR cell surface expression level. *ABCC4* appears to act as a hepatic transporter that determines the efflux of cAMP. Upon the loss of *ABCC4*, the intracellular cAMP sensor Epac2 is activated, which directly activates small GTPase Rap1 and further downregulates PCSK9 protein expression, thus enhancing LDLR surface availability by preventing its lysosomal degradation. In general, we identified a key regulator of LDLR expression in hepatocytes, which offered novel insights into potential therapeutic targets for hypercholesterolemia and lipid-related disorders.

2. What is the pattern of expression of *Abcc4* in tissues and cells in both mice and humans ? Is *Abcc4* constitutively expressed in all cells and tissues ? What is known on the regulation of *Abcc4* expression ? Is the *Abcc4* gene regulated by cholesterol or lipids ? The authors should give more information on this transporter.

Answer: We appreciate the reviewer's suggestions and we have provided more information on this transporter *ABCC4* in our manuscript (line 143-150; line 160-168).

ABCC4, the top-ranking putative negative regulator of LDLR identified in our screens, is expressed in various tissues, including the liver, brain, kidney, colon, lung and placenta^{38,39}. Subcellularly, *ABCC4* is localized to the basolateral membrane of hepatocytes, also referred to as the sinusoidal surface, where it transports several conjugates across the basolateral membrane into the bloodstream⁴⁰⁻⁴². Moreover, the *ABCC4* transporter augments the potential for platelet activation by facilitating cAMP efflux, which in turn intensifies cAMP-dependent signal transduction to the nucleus^{43,44}. However, *ABCC4* expression is highly variable but cannot be fully attributed to known mechanisms. Several work conceived the concept that human *ABCC4* and rodent *Abcc4* were regulated by cholestasis and microRNA-124a and microRNA-506.

3. Is there any association between *ABCC4* SNPs and cholesterol levels and/or ASCVD in human ?

More importantly, what is the localization of *Abcc4* expression in hepatocytes ?

Answer: We appreciate the reviewer's suggestions which were helpful in improving our manuscript. According to the reviewer's suggestions, we have revised the manuscript extensively to explain this issue (line 143-148; line 275-288).

ABCC4, the top-ranking putative negative regulator of LDLR identified in our screens, is expressed in various tissues, including the liver, brain, kidney, colon, lung and placenta^{38,39}. Subcellularly, *ABCC4* is localized to the basolateral membrane of hepatocytes, also referred

to as the sinusoidal surface, where it transports several conjugates across the basolateral membrane into the bloodstream⁴⁰⁻⁴².

Our findings demonstrate that either genetic ablation (**Fig. 3**) or pharmacological inhibition of hepatic ABCC4 (**Fig. 4**) effectively increases hepatic LDLR protein levels and lowers plasma LDL-C levels by approximately 40% in mice. Human and mouse ABCC4 proteins are highly conserved and share 100% sequence identity, which suggests potentially conserved function across species^{39,41,42}. To investigate the relationship between the ABCC4 gene locus and plasma LDL-C levels in larger cohorts, we analyzed publicly available datasets from the UK Biobank and the Global Lipid Genetics Consortium^{68,69}. We identified additional variants at the ABCC4 gene locus that were associated with plasma lipids and cardiovascular risks (**Extended Data Fig. 5a**). Using public datasets from the STARNET study (Stockholm-Tartu Atherosclerosis Reverse Networks Engineering Task)⁷⁰, we also found that ABCC4 was part of a liver gene expression coregulatory network module significantly associated with coronary artery disease and cardiometabolic disease outcomes (**Extended Data Fig. 5b**).

Extended Data Fig. 5 | The human ABCC4 gene is linked to plasma lipids and cardiovascular diseases. a, Table summarizing the variants in the ABCC4 gene locus that are associated with plasma lipids and cardiovascular risks. **b**, Human gene co-regulatory network of STARNET data for ABCC4 module membership and cardiometabolic phenotype associations.

4. Figure 1. Although the MAGeCK strategy to identify potential targets controlling LDLR expression is interesting, the Reviewer has some doubts on its efficiency since major genes known to control LDLR expression such as PCSK9 and MYLIP seem to not be statistically identified (Supplemental Tables). However, the authors indicated in the Discussion section (line 308) that “In addition, a subset of negative hits such as PCSK9 and MYLIP were exactly confirmed by using our individual sgRNAs”. This point is not clear and must be detailed by the authors and data must be shown.

Answer: We are grateful to the reviewer for providing such an excellent advice. We have detailed the point and explained this limitation in the “Discussion” part (line 479-491).

In this study, we performed a genome-scale screening by introducing a library targeting over 18,000 protein-coding genes into hepatocytes to identify genes whose deficiency culminated in the LDLR^{high} phenotype. The top 10% of LDLR^{high}-expressing cells were sorted using an anti-LDLR antibody. Among the scoring genes analyzed by MAGeCK, ABCC4 was identified as a gene of significant importance based on our data (**Fig. 1c**). Knocking out ABCC4 exerted a regulatory effect on surface LDLR abundance in hepatocytes, thereby promoting LDL uptake. In addition to novel modifiers, LDLR^{53,55}, SREBF2⁸⁰ and MBTPS1⁸³ (membrane-bound transcription factor peptidase, site 1) were identified as positive hits in our screening data, although they did not exhibit high ranking or statistical significance.

While our screening identified novel modifiers for surface LDLR protein expression, it failed to prioritize the effects of some canonical regulators. A general reason for this issue is the not optimal signal-to-noise ratio of our screening. Specifically, a reason for missing MYLIP or PCSK9 is the optimization of our screening towards the discovery of loss-of-function effects. Additionally, our screening primarily focused on genes that induce substantial transcriptional changes under homeostatic conditions, but post-transcriptional regulation also plays an important role in determining protein levels. Further studies should aim to compare the regulatory effects of transcriptional and post-transcriptional mechanisms. Furthermore, the observed variability in effects depending on the specific sgRNAs used indicates that additional research is needed to systemically characterize the switch performance of each designed sgRNA.

5. Figure 5. In addition, invalidation of *Abcc4* in AML12 cells was accompanied by a modification of several biological pathways by GSEA with none on cholesterol metabolism suggesting that *Abcc4* did not impact cholesterol uptake and metabolism in hepatocytes. The authors must discuss this discrepancy with their proposed mechanism.

Answer: We thank for the excellent comments which we believe can improve the manuscript. With regard to this question, we have again detailed the concept in the “Result” section (line 185-195; line 301-314).

Due to the undesirable weakness of RNA-seq technology, we only elucidate the most significant pathway involved the lipid metabolism process and downstream signals upon *Abcc4* deficiency, ignoring some insignificant information. Indeed, based on prior in vitro experiments, knockout of *Abcc4* did not impact cholesterol contents and sterol sensing in hepatocytes (**Extended Data Fig. 2h and 2i**).

We identified 940 differentially expressed genes (DEGs) (72.87%) that were significantly upregulated and 350 genes that were downregulated (27.13%) in AML12 cells, using a *P*-value threshold of < 0.05 and a fold change > 2.0 as the cutoff between *Abcc4*-KO and Control-KO (**Data S5**). Moreover, an in-depth analysis of these DEGs revealed significant enrichment in pathways including the cAMP signaling pathway, Rap1 signaling pathway, cGMP-PKG signaling pathway, Phospholipase D signaling pathway, insulin signaling pathway, insulin resistance, Glucagon signaling pathway and PPAR signaling pathway, which were closely linked to lipid metabolism, as supported by previous studies (**Fig. 5c**)^{45,71}. Gene set enrichment analysis (GSEA) demonstrated that *Abcc4* deficiency simultaneously activated a broad spectrum of genes associated with the cAMP signaling pathway (**Fig. 5d**), Rap1 signaling pathway (**Fig. 5e**). Notably, the knockout of *Abcc4* significantly altered the expression profiles of genes involved in the lipid metabolism process and the insulin signaling pathway (**Fig. 5f**). Meanwhile, a comprehensive transcriptome analysis of AML12 cells treated with DMSO or Ceefourin-1 revealed strikingly consistent gene expression profiles (**Extended Data Fig. 6**). Aligned with our in vivo findings, inhibition of ABCC4 markedly affected insulin-related signaling (**Extend Data Fig. 4g**).

6. Figure 2. Although the impact of *Abcc4* on the cell surface expression of LDLR and the LDL uptake is convincing, the authors however failed to prove that those effects led to an increased cholesterol content into the cell. Then, intracellular content should be quantified and the expression of cholesterol-dependent genes including SREBP2 and HMGCoAR should be analyzed.

Answer: We thank for the excellent comments and we have improved this concept in **Extended Data Fig.2** of the “Result” Section (line 185-203).

Cellular cholesterol level is essential for cell growth and maintenance while cholesterol uptake, synthesis and metabolism, is largely regulated by sterol regulatory element-binding protein 2 (SREBP2) and 3-hydroxy-3-methylglutaryl-CoA reductase (HMGCR)^{52,53}. We then evaluated cellular cholesterol contents in *Abcc4*-deficient AML12 cells by ELISA method and found that ABCC4 barely exerted effect on cellular cholesterol level (**Extend Data Fig. 2h**). Meanwhile, the loss of *Abcc4* did not influence SREBP2-dependent gene expression (*Srebp2* and *Hmgcr*), as evidence by RT-qPCR analysis (**Extend Data Fig. 2i**). As seen in Fig. 2g-2j, knockout of *Abcc4* in hepatocytes had significant effect on cell surface LDLR expression and LDL endocytosis, with no effect on sterol sensing (**Extend Data Fig. 2c, 2f, 2h and 2i**). Considering that the uptake of LDL by the LDLR results in its lysosomal degradation, we evaluated the degradation of Dil-LDL (after 4h)^{54,55} in cells deficient of ABCC4. Knockout of *Abcc4* in AML12 cells promoted the uptake of Dil-LDL, as well as the degradation of Dil-LDL (**Extend Data Fig. 2j**), which likely accounts for the minimal impact of ABCC4 on sterol sensing and cholesterol synthesis (LDLR, SREBP2 and HMGCR). Taken together, these data indicated that ABCC4 blockade resulted in increased hepatic LDLR protein levels on cell surface and enhanced lipoprotein uptake independent of LDLR gene transcription.

7. Figure 3-4. In vivo experiments in mice conducted in this study are intriguing. First of all, it is very surprising that a high-fat diet and not a high-cholesterol diet was used in these experiments to validate the role of Abcc4 on cholesterol uptake by the liver. What is the composition of the HFD used? In addition, why the authors did not use a more appropriate model for hypercholesterolemia such as ApoE KO mice (the use of Ldlr KO mice being not possible here)? This mouse model would have been very useful for studying the impact of Abcc4 depletion on atherosclerosis development. Finally, the authors must explain why they selected the ApoE/hAAT promoter for the expression of RNAi targeting Abcc4 “specifically in hepatocyte”? The use of largely used promoters such as Albumin or TBG would have been more appropriate. The authors must discuss this point. Was the same promoter used for the expression of control GFP_RNAis?

Answer: We are grateful to the reviewer for the constructive suggestion. We have revised this point in the “Methods” section (line 596-601).

Mice were fed with either a normal chow diet (NCD) or 60% high fat diet (HFD) for 4 weeks, 60% HFD (5.24 kcal/g with 20% energy derived from protein, 60% from fat, and 20% from carbohydrate; D12492; Research Diets, Inc., New Brunswick, New Jersey). A rodent diet containing 60 kcal% fat is typically used for studies related to obesity, metabolic disorders, and lipid-related disorders, as it closely mimics high-cholesterol diets found in human populations¹⁰¹⁻¹⁰³.

We appreciate the reviewer’s insightful comment regarding the choice of animal model. We agreed that *ApoE* knockout (*ApoE*^{-/-}) mice are a well-established model for studying severe hypercholesterolemia and atherosclerosis. However, we deliberately chose to use wild-type (WT) mice in combination with a 60% HFD diet (5.24 kcal/g with 20% energy derived from protein, 60% from fat, and 20% from carbohydrate; D12492; Research Diets, Inc., New Brunswick, New Jersey) to mimic the early stages of diet-induced metabolic syndrome and moderate hyperlipidemia in human populations¹⁰¹⁻¹⁰³. Moreover, using WT mice allowed us to evaluate the direct impact of dietary intervention on lipid metabolism without the confounding effects of genetic knockout. We acknowledge that the *ApoE*^{-/-} mice model could be valuable for future studies aiming to investigate advanced stages of hypercholesterolemia and atherosclerosis.

Lastly, we thank the reviewer for raising this important point and understand that promoters

like Albumin and TBG are also commonly used for liver-specific gene expression. In this study, we chose recombinant AAV serotype-8 expressing the ABCC4 RNAi under the liver-specific apolipoprotein E/human alpha 1-antitrypsin promoter [AAV8-ApoE/hAAT; 2×10^{11} viral genomes (VG)], which were administered through tail vein injection (Genechem, Shanghai, China). The ApoE/hAAT promoter is known for its strong and liver-specific expression, which has been demonstrated in previous studies to provide effective gene regulation specifically within hepatocytes^{56,57,104}. We and technicians believe that this promoter is particularly well-suited for models where a hepatocyte-specific expression pattern is required for investigating liver-specific gene functions, especially when combined with RNA interference strategies. The same promoter absolutely was used for the expression of control GFP_RNAi. We have revised the manuscript to include this explanation and provide more clarity regarding the rationale behind our choice of the ApoE/hAAT promoter in the “Methods” section (line 609-619).

8. Figure 3. Circulating cholesterol is mostly carried by HDL and not LDL in the mouse model and the diet used in this study. This very likely explains why the authors observed no significant changes in plasma total cholesterol levels between AAV8_GFP_RNAi controls and AAV8_Abcc4_RNAi mice. Isolation of lipoproteins would help to determine if cholesterol associated to LDL (LDL-C) was reduced as expected upon hepatic *Abcc4* deficiency. Surprisingly LDL-C quantification (although using a questionable assay) was performed in mice treated with Ceefourin-1 (Figure 4g) and the Reviewer wonders why this was not done in *Abcc4* deficient mice. A similar comment can be made with Oil Red O (ORO) staining in the liver (Figure 4h). However, a direct quantification of cholesterol in the liver would be more appropriate since ORO staining reflects both cholesterol and triglyceride accumulation in the liver.

Answer: We thank the reviewer again for these helpful suggestions, which have substantially improved the manuscript. We have measured liver LDL-C and TC contents in the “Result” section (line 218-229; line 250-274).

In our study, we treated Wild-Type (WT) C57BL/6 mice with adeno-associated virus type 8 (AAV8) in which small pieces of interfering double-stranded RNA targeting *Abcc4* (*Abcc4*_RNAi) was driven by the ApoE/hAAT promoter that decreases the transcription of *Abcc4* gene in the liver. Reduced hepatic ABCC4 remarkably upregulated liver tissue membrane LDLR protein expression levels and accordingly decreased serum LDL-C levels (**Fig. 3e-3g**), without significant changes in total cholesterol (TC) levels (**Extended Data Fig. 3c**). Indeed, we agreed that LDL-C only account for the minimal part circulating cholesterol in C57BL/6 mice, which explained the lack of significant differences in TC levels between the AAV8_GFP_RNAi controls and AAV8_Abcc4_RNAi groups under a normal-chow-diet condition. Additionally, we measured liver LDL-C and TC contents in mice and concordantly illustrated that there were no significances in liver LDL-C and TC contents in the AAV_Abcc4_RNAi group (**Fig. 3h and Extended Data Fig. 3c**). These data suggested that ABCC4 probably only regulates LDL cholesterol through LDLR, not total cholesterol.

In Fig. 4 of the “Results” section, we further evaluated the effect of pharmacological inhibition of ABCC4 on the clearance of circulating cholesterol in mice fed with a normal-chow diet (NCD) or high-fat diet (HFD). As we expected, Ceefourin-1 treatment exhibited significant effects on reduction of serum LDL-C levels by about 41.1% in mice of NCD group and 42.2% in mice of HFD group (**Fig. 4g**). Similar to the results in Fig. 3h, our data showed that pharmacologic inhibition of ABCC4 had little effect on liver LDL-C levels (**Fig 4h**). Additionally, we observed no significant differences in serum TC and liver TC levels in mice treated with Ceefourin-1 or DMSO (**Fig. 4i and 4j**). However, we intriguingly found that Ceefourin-1 greatly reduced liver TC levels in mice fed with a high-fat diet (**Fig. 4j**). The possible mechanism underlying this role of ABCC4 in response to a high-fat diet is that the liver is a major player in the maintenance of glucose and lipid homeostasis, where hyperglycemia alongside with insulin resistance are predominant manifestations of numerous metabolic abnormalities and diseases⁶⁴⁻⁶⁷.

As we mentioned in the “Discussion” section, future work will be required to investigate the relationship between ABCC4 gene and metabolic disorders such as insulin resistance, obesity, diabetes and hyperlipidemia. Moreover, it is essential to further validate this effect in hepatocyte-specific *Abcc4* knockout (KO) mice under metabolic disorder conditions. Such studies would allow us to better understand its role in glucose and lipid metabolism, as well as its potential contribution to the development of atherosclerotic lesions.

9. Figure 4. As a follow up of the previous comment, how do the authors explain the reduction of neutral lipid accumulation in the liver from mice upon treatment with the *Abcc4* inhibitor, Ceefourin-1? Based on the mechanism proposed by the authors for *Abcc4*, an increased ORO staining would be expected (increased LDLR-dependent uptake of cholesterol). These data questioned on the specificity of Ceefourin-1. The authors must clarify this point.

Answer: We thank the reviewer again for this insightful observation and have explained this point in the “Results” and “Discussion” sections (line 234-274; line 446-454).

Previous studies have shown that Ceefourin-1 is a highly specific inhibitor of *ABCC4*⁵⁹⁻⁶², blocking the transport of its substrates such as 6-mercaptopurine (6-MP), cAMP, D-Luciferin and estradiol-17 β glucuronide (E217 β G), with no detectable effect on its homologs MRP1, 2, 3 or 5. Ceefourin-1, utilized to disrupt *ABCC4* function, reduces platelet adhesion and thrombus formation by inhibiting cAMP efflux⁶¹. We found that Ceefourin-1 treatment exhibited significant effects on reduction of serum LDL-C levels by about 41.1% in mice of NCD group and 42.2% in mice of HFD group (**Fig. 4g**). Similar to the results in Fig. 3h, our data showed that pharmacologic inhibition of *ABCC4* had little effect on liver LDL-C levels (**Fig 4h**). Additionally, we observed no significant differences in serum TC and liver TC levels in mice treated with Ceefourin-1 or DMSO under a NCD condition (**Fig. 4i and 4j**).

Although previous in vitro data (**Extend Data Fig. 2c, 2f, 2h, 2i and 2j**) revealed that knockout of *Abcc4* in AML12 cells promoted the uptake of Dil-LDL, as well as the degradation of Dil-LDL, barely exerting effect on sterol sensing and cholesterol synthesis.

Intriguingly, we found that Ceefourin-1 greatly reduced liver TC levels in mice fed with a high-fat diet (**Fig. 4j**). H&E and Oil red staining actually showed that pharmacological inhibition of ABCC4 markedly improved lipid accumulation in liver tissues in comparison with controls treated with DMSO, especially in HFD group (**Fig. 4k**). In vivo models, the liver is a major player in the maintenance of glucose and lipid homeostasis, where hyperglycemia alongside with insulin resistance are predominant manifestations of numerous metabolic abnormalities and diseases⁶⁴⁻⁶⁷. Hence, we compared fasting blood glucose and fasting serum insulin, glucose tolerance, and insulin tolerance in mice treated with Ceefourin-1 or DMSO. As anticipated, HFD feeding significantly increased levels of fasting blood glucose and serum insulin in mice compared with NCD feeding (**Extend Data Fig. 4e**). Consistently, pharmacologic inhibition of ABCC4 by Ceefourin-1 improved glucose tolerance and ameliorated insulin resistance in HFD-fed mice, as indicated by the glucose tolerance test (GTT) (**Extend Data Fig. 4f**) and the insulin tolerance test (ITT) (**Extend Data Fig. 4g**). The precise mechanisms underlying the protective role of ABCC4 in response to metabolic disturbances induced by a high-fat diet remain unclear. One possibility is that the loss of ABCC4 function may modulate glucagon signaling, contributing to a coordinated regulatory effect on metabolic abnormalities and related diseases. Future work will be required to investigate the relationship between ABCC4 gene and metabolic disorders such as insulin resistance, obesity, diabetes and hyperlipidemia.

10. Figure 5C. Transcriptome analysis strongly suggests that biological pathways linked to diabetes and insulin resistance rather than cholesterol metabolism are modulated upon *Abcc4* deficiency. This could explain in part the reduced lipid accumulation in the liver upon HFD diet in mice treated with Ceefourin-1. Were insulin and glucose tolerance tests performed in mice ? What about fasting plasma insulin and glucose levels ? Plasma triglyceride levels ? This point is critical.

Answer: We thank the reviewer for this highly relevant and insightful comment. As noted, our transcriptome analysis in Fig. 5c, 5f indeed revealed a significant enrichment of pathways related to insulin signaling pathway, insulin resistance, and Glucagon signaling pathway in hepatocytes deficient of *Abcc4*. Hence, we did measure fasting blood glucose and serum insulin levels, as well as performed insulin and glucose tolerance tests in mice treated with Ceefourin-1 or DMSO (**Extended Data Fig. 4e-4g**) (line 262-269).

HFD feeding significantly increased levels of fasting blood glucose and serum insulin in mice compared with NCD feeding (**Extend Data Fig. 4e**). Consistently, pharmacologic inhibition of ABCC4 by Ceefourin-1 improved glucose tolerance and ameliorated insulin resistance in HFD-fed mice, as indicated by the glucose tolerance test (GTT) (**Extend Data Fig. 4f**) and the insulin tolerance test (ITT) (**Extend Data Fig. 4g**).

11. Figures 5i-k. Was secreted PCSK9 levels also reduced in AML12 cells treated with Abcc4 sgRNAs ?

Answer: We thank the reviewer for raising this important point. To address this, we have now conducted the quantification of secreted PCSK9 by ELISA methods in the culture supernatants of AML12 cells (**Fig. 5k and 5o**) and in mouse serum (**Extended Data Fig. 7**) (line 329-337).

As anticipated, we observed a corresponding decrease in both intracellular PCSK9 and secreted PCSK9 levels without significant changes in *Pcsk9* mRNA expression in hepatocytes lacking ABCC4 (**Fig. 5i-5l**). Importantly, similar reductions in both intracellular and secreted PCSK9 protein levels were found in AML12 cells under Ceefourin-1 treatment, with no significant changes in *Pcsk9* mRNA transcription levels (**Fig. 5m-5p**).

Moreover, in animal models, both liver-specific disruption and pharmacological inhibition of ABCC4 consistently resulted in decreased hepatic PCSK9 protein expressions and lower plasma PCSK9 levels compared to the control group (**Extend Data Fig. 7a-7e**).

12. Figure 2j and 6d. It would be very elegant to show that addition of recombinant PCSK9 abolishes the increased uptake of DiI-LDL in *Abcc4* deficient cells. This would reinforce the role of *Abcc4* in the LDLR-dependent uptake of cholesterol.

Answer: We thank the reviewer for this helpful suggestion and revised this point in the “Results” section (line 344-349).

To further explore the link between *ABCC4* and *PCSK9*, we next incubated AML12 cells with recombinant human *PCSK9* protein (rh*PCSK9*) stimulation (500 ng/ml). Notably, rh*PCSK9* treatment completely abolished the effect of *Abcc4* silencing on LDLR surface expression and LDL uptake (Fig. 5r and 5s), suggesting that *ABCC4* plays a critical role in modulating *PCSK9* protein levels and thereby influenced LDLR protein expressions and LDL-C clearance.

13. Figure 6b, line 260. “Interestingly, elevated LDLR surface abundance induced by *Abcc4* deficiency was abrogates in *Epac2* knockout AML12 cells”. This assertion is not true since no experiment was performed with a combined inhibition of both *Abcc4* and *Epac2* by sgRNA. As a whole, the authors failed to validate the requirement of the *Epac2* / *Rap1* pathway in the reduction of *PCSK9* expression and DiI-LDL operated upon *Abcc4* deficiency. This needs to be validated in experiments with both single and dual silencing of *Abcc4* and either *Epac2* or *Rap1*, respectively.

Answer: We are highly obliged to the reviewer for provision of the thoughtful suggestions and we have revised our “Results” section of manuscript accordingly (line 365-391).

It was widely accepted that the intracellular effects of cAMP are attributed to two primary receptors, the classical protein kinase A (PKA)/cAMP-dependent protein kinase and the

exchange protein directly activated by cAMP (EPAC)/cAMP-regulated guaninenucleotide exchange factors (GEFs)^{72,73}. Accumulating evidence indicates that EPAC proteins bind to cAMP with high affinity and activate small GTPases from the RAS superfamily, including RAP1 and RAP2⁷⁴, which is consistent with our transcriptome data analysis (**Fig. 5c-5f**). To determine whether changes in intracellular cAMP levels impact downstream Epac2/Rap1 signaling, we generated *Abcc4/Epac2* double-knockout (DKO) AML12 cells using two different sgRNAs targeting the gene (**Fig. 6a**). Interestingly, elevated LDLR surface abundance induced by *Abcc4* deficiency was abrogated in *Abcc4/Epac2* double-knockout AML12 cells (**Fig. 6b**). Moreover, incubated with Dil-LDL particles, *Abcc4/Epac2* DKO in cells impaired LDL uptake, in contrast to the positive effect observed with *Abcc4* deficiency (**Fig. 6c and 6d**). Notably, we also observed that lacking both *Epac2* and *Abcc4* in hepatocytes led to an increase in intracellular and secreted PCSK9 protein levels, without significant changes in *Pcsk9* mRNA expression (**Fig. 6e-6g**).

EPAC2 protein not only functions as an intracellular cAMP effector but also acts as a RAP1 activator, which was demonstrated by our RNA sequencing data (**Fig. 5**). Next, we investigated the role of Rap1 in regulating LDLR abundance. Double-knockout of *Abcc4/Rap1a* in AML12 cells depleted LDLR protein expression levels on cell membrane, as evidenced by immunoblotting data (**Fig. 6h and 6i**). In line with the decrease in LDLR surface abundance, *Abcc4/Rap1a* DKO in AML12 cells impeded LDL uptake by LDLR (**Fig. 6j and 6k**). Furthermore, double-knockout of *Abcc4/Rap1a* in hepatocytes significantly elevated intracellular and secreted PCSK9 protein levels without affecting its mRNA transcription (**Fig. 6l-6n**). Overall, these data highlight the ABCC4-cAMP-Epac2/Rap1a signaling pathway as a key regulator of surface LDLR availability in hepatocytes.

14. Discussion section, line 360. As mentioned above, this conclusion is not supported by data presented in this manuscript.

Answer: We are grateful to the reviewer for this suggestion. In summary, we have revised the major concerns mentioned above in the manuscript in close conformity with the requests of the reviewer. Hence, based on the previous revisions, we concluded that our study provides novel insights into the role of ABCC4 in regulating LDLR surface availability in hepatocytes dependent on cAMP-Epac2/Rap1a signaling, which contributes to the clearance for circulating LDL cholesterol. These findings position ABCC4 as a promising new therapeutic target for the intervention of hypercholesterolemia and atherosclerotic diseases.

15. Minor points. “stains” must be replaced by “statins” in several lines of the manuscript (lines 76, 357...)

Answer: We thank the reviewer for the excellent comments which we believe can improve the manuscript. We have now performed a thorough revision of the manuscript for grammar, spelling, and overall clarity.

Reviewer #2 (Remarks to the Author):

The manuscript of Chen et al. describes the discovery of the protein ABCC4/MRP4 as a novel modulator of cell surface low density lipoprotein (LDL) receptor (LDLR) in AML12 hepatocytes. The authors provide evidence that intracellular cAMP levels regulate PCSK9 levels, which regulate cell surface LDLR. The authors supported this data with animal work and work with a specific ABCC4 inhibitor ceefourin-1. Overall I find the work of good quality and conclusions are well supported by the results. Spelling and grammar are poor, with lots of mistakes in the text. The biggest problem is that this pathway has already been published and cited (reference 43), though the authors fail to properly present this paper. While I find the work with ceefourin-1 novel, overall this paper lacks novelty. My comments are below.

Answer: We appreciate the reviewer’s comments, which have helped us improve the clarity, rigor, and context of our manuscript. Detailed responses to each point are provided below.

1. The authors should describe a little about the proteins that were discovered in the initial screen, in addition to MDR4/ABCC4. There is certainly no cohesion in the proteins or pathways identified.

Answer: We thank the reviewer for this valuable suggestion and revised this point in the “Results” section (line 131-150).

By applying the MAGeCK (model-based analysis of genome-wide CRISPR-Cas9 knockout)^{29,30} computational algorithm to compare LDLR^{high} subpopulations and the unsorted populations, we ranked the genes in the two dependent screens (**Fig. 1b and 1c; Data S1 and S2**). At the conclusion of the screens, barplots showed the log₂ fold change (Log FC) of the top-scoring genes, comparing the LDLR^{high} subpopulation to the unsorted cell population (**Extended Data Fig. 1b and 1c**). Notably, by comparing the LDLR^{high} subpopulation with the unsorted population, we discovered previously reported modulators (UROD³¹, CHP1³¹, FASN³², TMEM251³³, CDC42^{34,35}, REPS2^{36,37}), as well as several novel candidate negative regulators for LDLR. Based on two independent screen replicates (M1 and M2), we found 68 gene hits from the M1 library that were also enriched as significant hits in the independent M2 library screen. These hits were subsequently selected for further analysis (**Fig. 1d; Data S3 and S4**).

ABCC4, the top-ranking putative negative regulator of LDLR identified in our screens, is expressed in various tissues, including the liver, brain, kidney, colon, lung and placenta^{38,39}. Subcellularly, ABCC4 is localized to the basolateral membrane of hepatocytes, also referred to as the sinusoidal surface, where it transports several conjugates across the basolateral membrane into the bloodstream⁴⁰⁻⁴². Moreover, the ABCC4 transporter augments the potential for platelet activation by facilitating cAMP efflux, which in turn intensifies cAMP-dependent signal transduction to the nucleus^{43,44}.

2. Line 174-175: What are the levels of LDL-C in this mouse model? The likely reason that the effect on total cholesterol is not significant is because, LDL-C makes up a small proportion of the total cholesterol.

Answer: We are grateful to reviewer for this advice. As the reviewer pointed out, we treated Wild-Type (WT) C57BL/6 mice with adeno-associated virus type 8 (AAV8) in which small pieces of interfering double-stranded RNA targeting *Abcc4* (*Abcc4*_RNAi) was driven by the ApoE/hAAT promoter^{56,57} that decreases the transcription of *Abcc4* gene in the liver (n=6, at a dose of 2×10^{11} viral genomes per animal) (**Fig. 3a; Extend Data Fig. 3a**) (line 209-229). Reduced hepatic ABCC4 remarkably upregulated liver tissue membrane LDLR protein expression levels and accordingly decreased serum LDL-C levels (**Fig. 3e-3g**), without significant changes in plasma TC levels (**Extended Data Fig. 3c**).

Indeed, we agreed that LDL-C only account for the minimal part circulating cholesterol in C57BL/6 mice, which explained the lack of significant differences in TC levels between the AAV8_GFP_RNAi controls and AAV8_Abcc4_RNAi groups under a normal-chow-diet condition. Additionally, we measured liver LDL-C and TC contents in mice and concordantly illustrated that there were no significant differences on liver LDL-C and TC contents in the AAV_Abcc4_RNAi group (**Fig. 3h and Extended Data Fig. 3c**). These data suggested that ABCC4 probably only regulates LDL cholesterol through LDLR, not total cholesterol.

3. Line 183: Did the authors mean cytosolic? If ABCC4 is a cAMP transporter, I assumed that it was externalizing cAMP.

Answer: We thank the reviewer for this valuable suggestion and revised this point in the “Result” section (line 234-238).

Previous studies have shown that Ceefourin-1 is a highly specific inhibitor of ABCC4⁵⁹⁻⁶², blocking the transport of its substrates such as 6-mercaptopurine (6-MP), cAMP, D-Luciferin and estradiol-17β glucuronide (E217βG), with no detectable effect on its homologs MRP1, 2, 3 or 5. Ceefourin-1, utilized to disrupt ABCC4 function, reduces platelet adhesion and thrombus formation by inhibiting cAMP efflux⁶¹.

4. Line 221-222: As a philosophical question, if ABCC4 knockout causes a wide range of alterations in gene expression, then would targeting ABCC4 as a therapeutic treatment be sensible?

Answer: We appreciate the reviewer's thought-provoking question. It is true that genetic knockout of *Abcc4* induces a broad set of transcriptional changes, as seen in our RNA-seq data. In our study, Ceefourin-1, a selective ABCC4 inhibitor, demonstrated a favorable effect on hepatic LDLR regulation and lipid metabolism without toxicity in vitro and in vivo model. This suggests that targeted inhibition of ABCC4 activity in hepatocytes could achieve beneficial outcomes. Nevertheless, we fully agree that careful therapeutic window assessment and tissue specificity would be essential considerations in future translational development of ABCC4-targeted therapies. We have now considered this important point in the revised Discussion section to clarify the limitation of current research (line 490-500).

While our findings unexpectedly revealed that ABCC4 inhibition significantly reduced liver TC levels and improved hepatic insulin resistance in the HFD model, it is essential to further validate this effect in hepatocyte-specific *Abcc4* knockout (KO) mice under metabolic disorder conditions. Such studies would allow us to better understand its role in glucose and lipid metabolism, as well as its potential contribution to the development of atherosclerotic lesions. Additionally, although our study identified ABCC4 inhibitor as a potential therapeutic target for hyperlipidemia, we did not provide evidence for its synergistic effect with statins or PCSK9 inhibitors. More studies are needed to explore and confirm the potential synergy between ABCC4 inhibition and other lipid-lowering therapies.

In conclusion, our work opens new perspectives for optimizing lipid-lowering therapy (LLT) in patients with high cardiovascular risk factors using ABCC4 inhibitors.

5. Line 246-248: The authors have not proven a correlation between cAMP levels and ABCC4. They are coincident. It is possible that ABCC4 has other functions besides cAMP transport, or that ceefourin-1 has other targets or effects. While I agree that the evidence has suggested an effect of ABCC4 on PCSK9, there is no evidence to support the effect of cAMP on PCSK9. In no way has the evidence proven anything, rather the evidence is supportive of a model where...

Answer: We thank the reviewer for the excellent comments which we believe can improve the manuscript. Indeed, previous literature have stated the role and function of ABCC4 in various physiological process, which are cited in our manuscript (line 143-153; line 160-165).

ABCC4 is a member of the superfamily of ATP-binding cassette (ABC) transporters, which mediates ATP-dependent efflux of a diverse range of endogenous and exogenous substrates. Multiple downstream signals or substrates of ABCC4 in mammals have been identified, including the classic second messengers cAMP and cGMP^{48,49}, the eicosanoids PGE1/PGE2⁵⁰, and the bile acids Cholyltaurine⁵¹. ABCC4 transporter augments the potential for platelet activation by facilitating cAMP efflux, which in turn intensifies cAMP-dependent signal transduction to the nucleus^{43,44}. Additionally, cAMP and its downstream signaling pathways act as the regulators of PCSK9 homeostasis and LOX-1 expression^{45,46}. In addition to cAMP,

ABCC4 transports prostaglandins like PGE₂, out of the cell, where they bind to receptors and modulate LDL-C levels⁴⁷.

In our study, we have clarified the relationship between cAMP levels and ABCC4 by RNA-seq and cAMP ELISA assays (line 292-320). A comprehensive transcriptome analysis of AML12 cells illustrated that differentially expressed genes (DEGs) between *Abcc4*-KO and Control-KO were significantly enriched in pathways including the cAMP signaling pathway, Rap1 signaling pathway, cGMP-PKG signaling pathway, Phospholipase D signaling pathway, insulin signaling pathway, insulin resistance, Glucagon signaling pathway and PPAR signaling pathway, which were closely linked to lipid metabolism, as supported by previous studies (**Fig. 5c**)^{45,71}. Gene set enrichment analysis (GSEA) demonstrated that *Abcc4* deficiency simultaneously activated a broad spectrum of genes associated with the cAMP signaling pathway (**Fig. 5d**), Rap1 signaling pathway (**Fig. 5e**). Similarly, AML12 cells treated with Ceefourin-1 led to the activation of cAMP signaling pathway (**Extended Data Fig. 6**).

It is obvious that *Abcc4*-knockout in AML12 cells effectively inhibited cAMP efflux and promoted intracellular cAMP signaling pathway, as evidenced by RNA-seq data. We further validated these findings by measuring both intracellular and extracellular cAMP levels in vitro using ELISA (**Fig. 5g**). Similarly, alterations of cAMP distribution were also observed in AML12 cells treated with the ABCC4 inhibitor Ceefourin-1 (**Fig. 5h**).

Additionally, previous researches have implicated that cAMP and its downstream signaling pathways act as the regulators of PCSK9 homeostasis and LOX-1 expression^{45,46}, as well as glucagon signaling in PCSK9 regulation⁷¹. As anticipated, we observed a corresponding decrease in both intracellular PCSK9 and secreted PCSK9 levels without significant changes in *Pcsk9* mRNA expression in hepatocytes lacking ABCC4 (**Fig. 5i-5l**). Importantly, similar reductions in both intracellular and secreted PCSK9 protein levels were found in AML12 cells under Ceefourin-1 treatment, with no significant changes in *Pcsk9* mRNA transcription levels (**Fig. 5m-5p**) (line 329-334).

In view of our transcriptome data analysis, *Rapgef4* (*Epac2*) and *Rap1a* play significant roles in AML12 cells deficient of *Abcc4*, we further determine whether changes in intracellular cAMP levels impact downstream Epac2/Rap1 signaling in regulating PCSK9 and LDLR expression (line 371-391).

Accumulating evidence indicates that EPAC proteins bind to cAMP with high affinity and activate small GTPases from the RAS superfamily, including RAP1 and RAP2⁷⁴, which is consistent with our transcriptome data analysis (**Fig. 5c-5f**). To determine whether changes in intracellular cAMP levels impact downstream Epac2/Rap1 signaling, we generated *Abcc4/Epac2* double-knockout (DKO) AML12 cells using two different sgRNAs targeting the gene (**Fig. 6a**). Interestingly, elevated LDLR surface abundance induced by *Abcc4* deficiency was abrogated in *Abcc4/Epac2* double-knockout AML12 cells (**Fig. 6b**). Moreover, incubated with Dil-LDL particles, *Abcc4/Epac2* DKO in cells impaired LDL uptake, in contrast to the positive effect observed with *Abcc4* deficiency (**Fig. 6c and 6d**). Notably, we also observed that lacking both *Epac2* and *Abcc4* in hepatocytes led to an increase in intracellular and secreted PCSK9 protein levels, without significant changes in *Pcsk9* mRNA expression (**Fig. 6e-6g**). EPAC2 protein not only functions as an intracellular

cAMP effector but also acts as a RAP1 activator, which was demonstrated by our RNA sequencing data (**Fig. 5**). Next, we investigated the role of Rap1 in regulating LDLR abundance. Double-knockout of *Abcc4/Rap1a* in AML12 cells depleted LDLR protein expression levels on cell membrane, as evidenced by immunoblotting data (**Fig. 6h and 6i**). In line with the decrease in LDLR surface abundance, *Abcc4/Rap1a* DKO in AML12 cells impeded LDL uptake by LDLR (**Fig. 6j and 6k**). Furthermore, double-knockout of *Abcc4/Rap1a* in hepatocytes significantly elevated intracellular and secreted PCSK9 protein levels without affecting its mRNA transcription (**Fig. 6l-6n**). Overall, these data highlight the ABCC4-cAMP-Epac2/Rap1a signaling pathway as a key regulator of surface LDLR availability in hepatocytes.

6. Line 263-275: The authors have provided good evidence to support the role cAMP and Epac in affecting LDLR surface presentation. The authors postulate that PCSK9 plays a role. As PCSK9 mRNA is not altered, but protein levels are affected, what is the hypothesis then? Impaired trafficking of PCSK9 so that it is not secreted? Phosphorylation of PCSK9?

Answer: We are grateful to the reviewer for excellent advice and revised the “Results” section in our manuscript (line 338-345).

Given that *Abcc4* deficiency did not alter *Pcsk9* mRNA levels (**Fig. 5l**), yet resulted in reduced intracellular and secreted PCSK9 levels (**Fig. 5i-5k**), these finding suggest that ABCC4 exerts a posttranscriptional regulatory effect on PCSK9 expression. Therefore, to assess whether ABCC4 influences PCSK9 protein stability, we treated cells with cycloheximide (30 mins and 90 mins) to inhibit protein synthesis. This treatment (30 mins) led to a marked reduction in PCSK9 protein levels in *Abcc4*-KO cells, suggesting increased degradation (**Fig. 5q**).

7. A published paper from 2018 has already established that Epac and Rap-1 affect PCSK9 protein levels (<https://www.ahajournals.org/doi/10.1161/CIRCRESAHA.118.313648>). This is reported as reference 43. In this paper, the authors suggest that glucagon signaling affects PCSK9 levels. I believe that the authors have simply tied into this already established pathway. This paper must be properly described in the text. Furthermore, the authors should perform experiments with glucagon to either prove or disprove that the findings of this paper

support previously published work. Furthermore, another paper has already established that MRP4/ABCC4 is tied to obesity and type 2 diabetes (<https://pmc.ncbi.nlm.nih.gov/articles/PMC8628361/>). So, this paper lacks novelty, except with the treatment with ceefourin-1.

Answer: We appreciate the reviewer's suggestions which are helpful to improve our manuscript. Indeed, a published paper has already established the role of Epac2/Rap1 signaling in PCSK9 regulation. Hence, we have supplemented this point in our manuscript according to the reviewer's suggestion (line 351-360).

Consistent with earlier studies implicating glucagon signaling in PCSK9 regulation^{45,71}, our RNA-seq analysis of *Abcc4*-KO AML12 cells revealed significant enrichment of genes involved in this pathway (**Fig. 5c**). In the next set of experiments, we indeed found that glucagon (Glu) treatment significantly decreased both intracellular and secreted PCSK9 levels, mirroring the effect of Ceefourin-1. Obviously, combined treatment with glucagon and Ceefourin-1, it exerted a synthetic effect on PCSK9 regulation, reinforcing the mechanistic role of ABCC4 in the target gene (**Extend Data Fig. 8a and 8b**). These results suggest that ABCC4 inhibitor could be a promising therapeutic strategy for hypercholesterolemia and lipid-associated metabolic disorders.

Moreover, we compared fasting blood glucose and serum insulin levels, glucose tolerance, and insulin tolerance in mice treated with Ceefourin-1 or DMSO (line 256-269).

Intriguingly, we found that Ceefourin-1 greatly reduced liver TC levels in mice fed with a high-fat diet (**Fig. 4j**). H&E and Oil red staining indeed showed that pharmacological inhibition of ABCC4 markedly improved lipid accumulation in liver tissues in comparison with controls treated with DMSO, especially in HFD group (**Fig. 4k**). The liver is a major player in the maintenance of glucose and lipid homeostasis, where hyperglycemia alongside with insulin resistance are predominant manifestations of numerous metabolic abnormalities and diseases⁶⁴⁻⁶⁷. Hence, we compared fasting blood glucose and serum insulin levels, glucose tolerance, and insulin tolerance in mice treated with Ceefourin-1 or DMSO. As anticipated, HFD feeding significantly increased levels of fasting blood glucose and serum insulin levels in mice compared with NCD feeding (**Extend Data Fig. 4e**). Consistently, pharmacologic inhibition of ABCC4 by Ceefourin-1 improved glucose tolerance and ameliorated insulin resistance in HFD-fed mice, as indicated by the glucose tolerance test (GTT) (**Extend Data Fig. 4f**) and the insulin tolerance test (ITT) (**Extend Data Fig. 4g**). The precise mechanisms underlying the protective role of ABCC4 in response to metabolic disturbances induced by a high-fat diet remain unclear. One possibility is that the loss of ABCC4 function may modulate glucagon signaling, contributing to a coordinated regulatory effect on metabolic abnormalities and related diseases. Future work will be required to investigate the

relationship between *ABCC4* gene and metabolic disorders such as insulin resistance, obesity, diabetes and hyperlipidemia.

8. Does *ABCC4* affect IDOL levels? Is LDLR ubiquitinated? IDOL/MYLIIP needs to be ruled out.

Answer: We appreciate the reviewer's suggestions and revised the "Results" section in our manuscript (line 160-165; line 169-203).

As expected, knockout of *Abcc4* increased the cell-surface LDLR abundance but not total LDLR protein levels (**Fig. 2c-2f; Extend Data Fig. 2a and 2b**). To test whether cell surface LDLR is upregulated by higher gene transcription levels, we compared *Ldlr*/*LDLR* mRNA levels between Control-KO (*Rosa26*-KO in murine cells and AAVS1-KO in human cells) and *Abcc4*/*ABCC4*-KO cells. We did not detect a significant difference in *Ldlr* mRNA expression between Control-KO and *Abcc4*-KO cells (**Extend Data Fig. 2c**). To test the generality of this regulation, we also observed similar results in human hepatocytes cell line, LO2 cells (**Fig. 2k-2o; Extend Data Fig. 2d-2g**). Considering that the uptake of LDL by the LDLR results in its lysosomal degradation, we evaluated the degradation of Dil-LDL (after 4h)^{54,55} in cells deficient of *Abcc4*. Knockout of *Abcc4* in AML12 cells promoted the uptake of Dil-LDL, as well as the degradation of Dil-LDL (**Extend Data Fig. 2j**), which likely accounts for the minimal impact of *ABCC4* on sterol sensing and cholesterol synthesis (LDLR, SREBP2 and HMGCR). From the work, we provides novel insights into the role of *ABCC4* in regulating LDLR surface availability by blocking PCSK9 protein expression, and thereby promoting its recycling back to the cell surface. Taken together, these data indicated that *ABCC4* blockade resulted in increased hepatic LDLR protein levels on cell surface and enhanced lipoprotein uptake independent of LDLR gene transcription or post-translational modification.

On the other hand, *ABCC4*, the top-ranking putative negative regulator of LDLR identified in our screens, is expressed in various tissues, including the liver, brain, kidney, colon, lung and placenta^{38,39}. Subcellularly, *ABCC4* is localized to the basolateral membrane of hepatocytes, also referred to as the sinusoidal surface, where it transports several conjugates across the basolateral membrane into the bloodstream⁴⁰⁻⁴². Moreover, *ABCC4* is a member of the superfamily of ATP-binding cassette (ABC) transporters, which mediates ATP-dependent

efflux of a diverse range of endogenous and exogenous substrates. Multiple downstream signals or substrates of ABCC4 in mammals have been identified, including the classic second messengers cAMP and cGMP^{48,49}, the eicosanoids PGE1/PGE2⁵⁰, and the bile acids Cholyltaurine⁵¹.

Hence, considering the two aspects of ABCC4's function and its effect on the cell-surface LDLR abundance but not total LDLR protein levels, we tend to think about the LDL-LDLR cycling pathway, not ubiquitination modification of LDLR.

Reviewer #3 (Remarks to the Author)

In this paper the authors performed genome wide screens in order to identify novel regulators of LDL receptor expression which could potentially be used as targets for LDL lowering therapies. Using this approach the authors focused on the top candidate which is the membrane transporter *Abcc4*. Using inhibitors and in vitro/in vivo studies they propose that *Abcc4* inhibition leads to upregulation of the LDLR gene via a pathway that includes cAMP, the cAMP sensor Epac, the small GTPase Rap1 and PCSK9.

Overall, the experiments are well designed and controlled but the main weakness of the paper is that it lacks sufficient mechanistic confirmation regarding the proposed signal transduction pathway that leads to the non-transcriptional inhibition of PCSK9. For instance no evidence is provided regarding the activation of the small GTPase Rap1 and the same applies to the other effectors of this pathway, including PCSK9. Based on the above, the data in this paper should be considered as preliminary. The authors should also check the manuscript thoroughly for grammar errors and typos.

Answer: We thank the reviewer for the excellent comments which we believe can improve the manuscript. And, we have elaborated mechanistic confirmation in detail underlying ABCC4's regulation in hepatic surface LDLR abundance in close conformity with the requests of the reviewer. The following summarized the revisions made to our manuscript in response to the comments (line 315-349; line 371-391).

Mechanistically, RNA seq analysis of transcriptome profiling in AML12 cells between *Abcc4*-KO and Control-KO groups was performed and illustrated that *Abcc4* deficiency simultaneously activated a broad spectrum of genes associated with the cAMP signaling pathway (**Fig. 5d**), Rap1 signaling pathway (**Fig. 5e**). Similarly, AML12 cells treated with Ceefourin-1 led to the activation of cAMP signaling pathway (**Extended Data Fig. 6**). It is obvious that *Abcc4*-knockout in AML12 cells effectively inhibited cAMP efflux and promoted intracellular cAMP signaling pathway, as evidenced by RNA-seq data. We further validated these findings by measuring both intracellular and extracellular cAMP levels in vitro using ELISA (**Fig. 5g**). Similarly, alterations of cAMP distribution were also observed in AML12 cells treated with the ABCC4 inhibitor Ceefourin-1 (**Fig. 5h**). Building on prior in vitro findings that *Abcc4*-KO in hepatocytes enhances LDL uptake by LDLR and accelerates its lysosomal degradation, without impacting sterol sensing and cholesterol synthesis (**Fig. 2g-2j; Extend Data Fig. 2c, 2f, 2h and 2i**), we hypothesized that inhibiting ABCC4 could amplify the effects of clinically validated therapies, potentially offering additive or synergistic benefits. As anticipated, we observed a corresponding decrease in both intracellular PCSK9 and secreted PCSK9 levels without significant changes in *Pcsk9* mRNA expression in

hepatocytes lacking *Abcc4* (**Fig. 5i-5l**). Importantly, similar reductions in both intracellular and secreted PCSK9 protein levels were found in AML12 cells under Ceefourin-1 treatment, with no significant changes in *Pcsk9* mRNA transcription levels (**Fig. 5m-5p**). Given that *Abcc4* deficiency did not alter *Pcsk9* mRNA levels (**Fig. 5l**), yet resulted in reduced intracellular and secreted PCSK9 levels (**Fig. 5i-5k**), these findings suggest that ABCC4 exerts a posttranscriptional regulatory effect on PCSK9 expression. Therefore, to assess whether ABCC4 influences PCSK9 protein stability, we treated cells with cycloheximide (30 mins and 90 mins) to inhibit protein synthesis. This treatment (30 mins) led to a marked reduction in PCSK9 protein levels in *Abcc4*-KO cells, suggesting increased degradation (**Fig. 5q**). To further explore the link between ABCC4 and PCSK9, we next incubated AML12 cells with recombinant human PCSK9 protein (rhPCSK9) stimulation (500 ng/ml). Notably, rhPCSK9 treatment completely abolished the effect of *Abcc4* silencing on LDLR surface expression and LDL uptake (**Fig. 5r and 5s**), suggesting that ABCC4 plays a critical role in modulating PCSK9 protein levels and thereby influenced LDLR protein expressions and LDL-C clearance. Additionally, previous researches have implicated that cAMP and its downstream signaling pathways act as the regulators of PCSK9 homeostasis and LOX-1 expression^{45,46}, as well as glucagon signaling in PCSK9 regulation⁷¹. In view of our transcriptome analysis, *Rapgef4* (*Epac2*) and *Rap1a* play significant roles in AML12 cells lacking *Abcc4*, we further determine whether changes in intracellular cAMP levels impact downstream Epac2/Rap1 signaling in regulating PCSK9 and LDLR expression (line 371-391).

Accumulating evidence indicates that EPAC proteins bind to cAMP with high affinity and activate small GTPases from the RAS superfamily, including RAP1 and RAP2⁷⁴, which is consistent with our transcriptome data analysis (**Fig. 5c-5f**). In the next sets of the experiments, we generated *Abcc4/Epac2* double-knockout (DKO) AML12 cells using two different sgRNAs targeting the gene (**Fig. 6a**). Interestingly, elevated LDLR surface abundance induced by *Abcc4* deficiency was abrogated in *Abcc4/Epac2* double-knockout AML12 cells (**Fig. 6b**). Moreover, incubated with Dil-LDL particles, *Abcc4/Epac2* DKO in cells impaired LDL uptake, in contrast to the positive effect observed with *Abcc4* deficiency (**Fig. 6c and 6d**). Notably, we also observed that lacking both *Epac2* and *Abcc4* in hepatocytes led to an increase in intracellular and secreted PCSK9 protein levels, without significant changes in *Pcsk9* mRNA expression (**Fig. 6e-6g**).

EPAC2 protein not only functions as an intracellular cAMP effector but also acts as a RAP1 activator, which was demonstrated by our RNA sequencing data (**Fig. 5**). Next, we investigated the role of Rap1 in regulating LDLR abundance. Double-knockout of *Abcc4/Rap1a* in AML12 cells depleted LDLR protein expression levels on cell membrane, as evidenced by immunoblotting data (**Fig. 6h and 6i**). In line with the decrease in LDLR surface abundance, *Abcc4/Rap1a* DKO in AML12 cells impeded LDL uptake by LDLR (**Fig. 6j and 6k**). Furthermore, double-knockout of *Abcc4/Rap1a* in hepatocytes significantly elevated intracellular and secreted PCSK9 protein levels without affecting its mRNA transcription (**Fig. 6l-6n**). Overall, these data highlight the ABCC4-cAMP-Epac2/Rap1a signaling pathway as a key regulator of surface LDLR availability in hepatocytes.

To summarize, mechanistically, the increased LDLR surface availability was driven by altered intracellular cAMP distribution and the activation of its downstream signals Epac2/Rap1a, upon *Abcc4* deficiency. Activation of cAMP signaling pathway due to *Abcc4* deficiency suppressed hepatic PCSK9 protein levels, thereby arresting LDLR lysosomal degradation and maintaining systematic LDL cholesterol homeostasis.

Moreover, we have performed a thorough revision of the manuscript for grammar, spelling, and overall clarity.

In summary, we have revised the manuscript in close conformity with the requests of the reviewers and editors. We hope that the revised articles can be published in *Communications Biology*. We thank the editors and reviewers for their excellent comments and we expect for your reply.

With best regards,

Sincerely yours,

Xiaoqing Wang, MD.

Xiaoqing_Wang@uestc.edu.cn

Dear reviewers,

On behalf of my colleagues and myself, I would like to resubmit a revision of our manuscript entitled “**ABCC4 impairs the clearance of plasma LDL cholesterol through suppressing LDLR abundance on hepatic surface**” for consideration of publication in *Communications Biology*. The following are our responses to the comments of the reviewers.

We thank reviewers for their careful and intensive reading and thoughtful comments on our manuscript. We have carefully taken their comments into consideration in preparing our revision, which has resulted in a paper that is clearer, more compelling. The following provides a summary of our responses to the reviewers' comments. Thanks for considering our manuscript.

Best wishes.

Responses to the comments of the reviewers

Reviewer #1 (Remarks to the Author):

The authors have provided a large amount of work in order to address the Reviewer's comments. The new version of the manuscript is more convincing and new data support the mechanism proposed by the authors. Then, most of critical points were clarified but some still need to be elucidated. To note that it was quite difficult to review the revised version of the manuscript without the line numbering. Because of the addition of numerous new data, the manuscript must be thoroughly checked before further editing process (few typos and spelling errors, check the numbering of Figures.....).

Answer: We thank the reviewer for the excellent comments which we believe can improve the manuscript. And, we have added the line numbering and thoroughly checked the manuscript in close conformity with the requests of the reviewer.

Major points.

1. Introduction. “Besides PCSK9 antibodies, current clinical therapies including statins and NPC1 (NPC Like Intracellular Cholesterol Transporter 1) inhibitors.....”. I guess that NPC1 must be replaced by NPC1L1.

Answer: We thank the reviewer for this thoughtful comment. We have checked the sentence in the Introduction section and replaced NPC1 (NPC Like Intracellular Cholesterol Transporter 1) with NPC1L1 (NPC Like Intracellular Cholesterol Transporter 1) (line 82-85). Besides PCSK9 antibodies, current clinical therapies including statins^{13,14} and NPC1L1 (NPC Like Intracellular Cholesterol Transporter 1) inhibitors¹⁵ also enhance LDLR expression on hepatocytes, thereby promoting serum LDL-C clearance.

2. Previous point#3. The new Extended Data Figure 5 is interesting but the legend must be further described. What is the meaning of the adjusted p value in a) ? Are those SNPs associated with plasma lipids (if yes, which ones ? LDL-C ?) or cardiovascular risk (how was defined this risk ?) ? Abbreviations used in b) must be detailed.

Answer: We appreciate the reviewer’s suggestions and we have provided more information about the Extend Data Figure 5 in our manuscript and in the legend (line 286-294).

To investigate the relationship between the ABCC4 gene locus and plasma LDL-C levels in larger cohorts, we analyzed publicly available datasets from the UK Biobank and the Global Lipid Genetics Consortium^{68,69}. We identified additional variants at the ABCC4 gene locus that were associated with plasma LDL cholesterol levels (**Extend Data Fig. 5a**). Using public datasets from the STARNET study (Stockholm-Tartu Atherosclerosis Reverse Networks Engineering Task)⁷⁰, we also found that ABCC4 was part of a liver gene expression coregulatory network module significantly associated with coronary artery disease and cardiometabolic disease outcomes (**Extend Data Fig. 5b**).

Extend Data Fig. 5 | The human ABCC4 gene is linked to plasma LDL-C levels and cardiovascular diseases.

a, Table summarizing the variants in the ABCC4 gene locus that are associated with plasma LDL-C levels. Statistical analysis was performed by Student’s test in [a]. **b**, Human gene co-regulatory network of STARNET data for ABCC4 module membership and cardiometabolic phenotype associations. MAM, free internal mammary artery; AOR, atherosclerotic aortic root; SF, subcutaneous fat; VAF, visceral abdominal fat; SKLM, skeletal muscle; LIV, liver; BMI, body mass index; P-Chol, plasma total cholesterol levels; fP-HDL-Chol, fasting plasma high-density lipoprotein (HDL) cholesterol levels; fP-LDL-Chol, fasting plasma LDL cholesterol levels; fP-TG, fasting plasma triglyceride levels; CRP, C-reactive protein; HbA1c, Hemoglobin A1C; CAD DEG, differentially expressed genes in coronary artery disease; SYNTAX, Synergy between Percutaneous Coronary Intervention with Taxus and Cardiac Surgery;

3. Previous point#4. “Specifically, a reason for missing MYLIP or PCSK9 is the optimization of our screening towards the discovery of loss-of-function effects.” This point is not clear for me. A loss-of function in PCSK9 is supposed to have similar effects than in ABCC4 (Reduced LDL-C and increased LDL-R in hepatocytes). Then PCSK9 should have been detected at a similar level than ABCC4. Is it correct ?

Answer: We appreciate the reviewer's suggestions which were helpful in improving our manuscript. According to the reviewer's suggestions, we have detailed the point and explained this limitation in the Discussion section (line 486-501).

In this study, we performed a genome-scale screening by introducing a library targeting over 18,000 protein-coding genes into hepatocytes to identify genes whose deficiency culminated in the LDLR^{high} phenotype. The top 10% of LDLR^{high}-expressing cells were sorted using an anti-LDLR antibody. Among the scoring genes analyzed by MAGeCK, ABCC4 was identified as the top-ranking gene based on our data (**Fig. 1c**).

While our screening identified novel modifiers for surface LDLR protein expression, it failed to prioritize the effects of some canonical regulators. A general reason for this issue is the not optimal signal-to-noise ratio of our screening. Technical noise and stochastic dropout are known issues in pooled screens, especially for genes with modest effects or variable knockout efficiency. It is possible that sgRNAs targeting PCSK9 or MYLIP were underrepresented or less effective in our system. In contrast, knockout of ABCC4 may have triggered a more robust or multi-level effect in our screening, and thus producing a more detectable phenotype. Additionally, our screening primarily focused on genes that induce substantial transcriptional changes under homeostatic conditions, but post-transcriptional regulation also plays an important role in determining protein levels. Further studies should aim to compare the regulatory effects of transcriptional and post-transcriptional mechanisms. Furthermore, the observed variability in effects depending on the specific sgRNAs used indicates that additional research is needed to systemically characterize the switch performance of each designed sgRNA.

4. Previous point#5. Once again, the answer to this question is not clear. What do the authors mean by “undesirable weakness of RNA-seq technology” ? The new data presented in (Extended Data Fig. 2h and 2i) are intriguing and seem not to support a role of ABCC4 in the expression of LDL-R at the cell surface. It is indeed unclear how an increased LDL uptake in AML12 cells is not accompanied by an elevated intracellular cholesterol content. Do the author envisage a compensatory pathway which might mask this effect (cholesterol export, synthesis of biliary acids.....) ? The authors must discuss this potential discrepancy. Indeed, the proposed explanation is poorly convincing “These data suggested that ABCC4 probably only regulates LDL cholesterol through LDLR, not total cholesterol”.

Other point, I do not understand the relevance to quantify LDL-C in the intracellular compartment as LDL are secreted by hepatocytes. Finally, the title of the Extended Data Fig. 2h must be corrected (“cholesterol” and not “cholesterol”).

Answer: We thank the reviewer for highlighting this important point and have elaborated on possible explanations in detail (line 185-204). The observation that *Abcc4*-deficient AML12 cells showed increased LDL uptake without a corresponding increase in cellular cholesterol contents (**Extend Data Fig. 2h**) suggests the existence of compensatory pathways for cholesterol export. LDL particles bind to an LDL receptor on the plasma membrane through receptor-mediated endocytosis. As the pH in the endosome decreases, LDLR undergoes characteristic changes of conformation. Then, LDLR recycles back to the hepatocyte surface, whereas LDL-C is transported to lysosomes for degradation. Considering that the uptake of

LDL by the LDLR results in its lysosomal degradation, we evaluated the degradation of Dil-LDL (after 4h)^{54,55} in cells deficient of ABCC4. Knockout of *Abcc4* in AML12 cells promoted the uptake of Dil-LDL, as well as the degradation of Dil-LDL (**Extend Data Fig. 2j**), which likely accounts for the minimal impact of ABCC4 on sterol sensing (LDLR, SREBP2 and HMGCR) and cellular cholesterol contents. Moreover, hepatocytes possess robust metabolic pathways for converting cholesterol into bile acids or oxysterols, which masked the effect of elevated LDL cholesterol influx. These data together indicated that ABCC4 blockade resulted in increased hepatic LDLR protein levels on cell surface and enhanced lipoprotein uptake independent of LDLR gene transcription.

Indeed, the gene set enrichment analysis (GSEA) in AML12 cells revealed *Abcc4*-KO significantly affected pathways related to insulin signaling and glucose metabolism, barely showing clear enrichment of canonical cholesterol metabolism pathways. A major component of our proposed mechanism involves ABCC4-cAMP-Epac2/Rap1a signaling pathway as a key regulator of surface LDLR availability in hepatocytes through modulation of PCSK9 expression, which governs lysosomal degradation of LDLR protein. This process does not necessarily involve broad transcriptional changes in classical cholesterol metabolism genes (HMGCR, SREBP2), which may explain their absence from enriched pathways. Additionally, hepatocytes maintain cholesterol levels through robust feedback loops. Therefore, modest increases in cholesterol uptake may not translate into detectable transcriptomic shifts in the short term, especially in vitro where compensatory efflux may buffer cellular cholesterol levels without triggering a transcriptional response. Our functional assays in vitro and in vivo revealed remarkable phenotypic effects of ABCC4 knockout or inhibition on LDLR surface availability in hepatocytes and plasma LDL-C clearance, despite the lack of transcriptional changes in cholesterol-related pathways.

Lastly, we agree that LDL particles themselves are secreted by the liver and that LDL-C, as a plasma lipoprotein fraction, is not found intracellularly in its native form. To clarify, our intention in **Extend Data Fig. 2h** was not to measure cellular LDL-C content, but rather to quantify cellular cholesterol content following increased LDL uptake via surface LDLR. We have revised the text in the figure legend and Results section to explicitly state that this assay measures cellular cholesterol content, which includes cholesterol derived from internalized LDL.

5. Previous point#8. The authors did not answer clearly to this question : “Surprisingly LDL-C quantification (although using a questionable assay) was performed in mice treated with Ceefourin-1 (Figure 4g) and the Reviewer wonders why this was not done in *Abcc4* deficient mice. A similar comment can be made with Oil Red O (ORO) staining in the liver (Figure 4h)”.

Answer: We thank the reviewer again for these helpful suggestions, which have substantially improved the manuscript. In response to the reviewer’s comment, we have measured LDL-C, TC and triglycerides (TG) levels in mice treated with AAV8_*Abcc4*_RNAi virus for disrupting ABCC4 expressions in liver, which were displayed in **Fig. 3** and **Extend Data Fig. 3** (line 213-218; line 222-232).

Genetic ablation of hepatic ABCC4 in vivo experiments, the omission of Oil red staining was primarily due to limited availability of liver tissues from the mice fed with a high-fat diet at the time of the initial analysis. Additionally, a direct quantification of triglyceride (TG) and cholesterol (TC) levels in the liver would be more appropriate for the mice fed with a normal-chow diet (NCD) using the commercial kits since Oil red staining reflects both cholesterol and triglyceride accumulation in the liver. Collectively, these new data reinforced the role of ABCC4 in regulating hepatic LDLR expression and plasma LDL cholesterol levels.

6. Previous points#9-10. The authors must conclude on the results from these interesting additional experiments. Does the increased ORO staining observed in liver from mice fed a HFD and treated with Ceefourin-1 result from an elevated glucose uptake and lipogenesis that could lead to a higher TG accumulation ? Once again, quantification of triglycerides in both plasma and liver would be helpful to elucidate this point. Other point : Bar legend from Extended Data Fig. 4e must be corrected (HFD was replaced by NCD in triangles).

Answer: We thank the reviewer for this thoughtful and mechanistically insightful question, which have substantially improved the manuscript. We have measured LDL-C, TC and triglycerides (TG) levels in mice administrated by ABCC4 inhibitor Ceefourin-1 in **Fig. 4** and **Extend Data Fig. 4** (line 255-266; line 273-276).

As anticipated, Ceefourin-1 treatment exhibited significant effects on reduction of serum LDL-C levels by about 41.1% in mice of NCD group and 42.2% in mice of HFD group (**Fig. 4g**), without significant differences in serum TC and TG levels (**Fig. 4h and 4i**). Based on our additional experiments, we intriguingly found that Ceefourin-1 treatment greatly reduced liver TC and TG levels in mice fed with a high-fat diet (**Fig. 4j and 4k**). Oil red staining indeed showed that pharmacological inhibition of ABCC4 markedly improved lipid accumulation in liver tissues compared to controls, especially in HFD group, along with enhanced LDLR expressions and plasma LDL-C clearance. The liver is a major player in the maintenance of glucose and lipid homeostasis, where hyperglycemia alongside with insulin resistance are predominant manifestations of numerous metabolic abnormalities and diseases⁶⁴⁻⁶⁷. Hence, we hypothesized that Ceefourin-1 treatment may alter glucose metabolism and de novo lipogenesis, thus alleviating hepatic lipid accumulation. Consistently, pharmacologic inhibition of ABCC4 by Ceefourin-1 improved glucose tolerance and ameliorated insulin resistance in HFD-fed mice, as indicated by the glucose tolerance test (GTT) (**Extend Data Fig. 4f**) and the insulin tolerance test (ITT) (**Extend Data Fig. 4g**). Our transcriptomic analysis indeed showed enrichment of pathways related to insulin signaling and glucose metabolism, suggesting that ABCC4 inhibition may have broader metabolic effects beyond cholesterol regulation. We have now addressed this hypothesis in the Discussion section of the revised manuscript, noting that the decreased cholesterol contents in mice fed with a high-fat diet under Ceefourin-1 treatment may be due in part to inhibited glucose-driven lipogenesis, rather than cholesterol accumulation alone. Future work will be required to investigate the relationship between ABCC4 gene and metabolic disorders such as insulin resistance, obesity, diabetes and hyperlipidemia.

Lastly, we have corrected the bar legend in Extend Data Fig. 4e.

Reviewer #2 (Remarks to the Author):

The authors have done a good job of addressing my concerns, with one exception.

Answer: We appreciate the reviewer's comments, which have helped us improve the clarity, rigor, and context of our manuscript. Detailed responses to the point are provided below.

1. There is some confusion about what LDL-C is. LDL-C is the cholesterol contained within LDL. It is a calculated measurement from the Friedewald formula. So, this is a plasma measurement of cholesterol contained within LDL lipoproteins. My confusion comes when the authors discuss "liver LDL-C" in Figure 3h, for example. If LDL is in the plasma, how can we measure liver LDL-C? This needs to be clarified.

Answer: We thank the reviewer for this valuable suggestion and revised this point in the Results and Methods sections (line 185-191; line 227-231; line 652-665).

Indeed, we agree entirely with the reviewer that LDL particles bind to LDL receptor on the plasma membrane, forming a receptor-ligand complex and internalizing in a clathrin-coated pit. After endocytosis, the LDL particles are then subjected to lysosomal degradation while LDLR recycles back to the surface.

In vivo models, we have measured serum LDL-C, TC and TG levels and liver TC and TG levels to well clarify the hepatic lipid deposition (**Fig 3 and 4; Extend Data Fig 3 and 4**). In vitro model, our intention in **Extend Data Fig. 2h** was not to measure cellular LDL-C contents, but rather to quantify cellular cholesterol contents following increased LDL uptake via surface LDLR. We have revised the text in the figure legend and Methods section to clearly state that the measurement reflects cholesterol contents extracted from liver tissues or AML12 cells, which includes cholesterol derived from internalized LDL.

Blood and tissue samples were collected immediately following the completion of the animal experiments for biochemical analysis. Serum was isolated by centrifugation of blood samples at 12,000× g for 10 min at 4°C. Liver tissue (approximately 100 mg) was homogenized and then centrifugated at 2500× g for 10 minutes at 4°C to collect the supernatant. The concentrations of total cholesterol (TC), low-density lipoprotein cholesterol (LDL-C) and triglycerides (TG) in serum and liver homogenates were measured using commercial kits (Nanjing Jiancheng Bioengineering Institute, Nanjing, China). Lipids were extracted from the AML12 cells by adding 0.2 mL of phosphate buffer for homogenization, followed by ultrasonic disruption in an ice water bath. The resulting homogenate was directly used to measure cellular cholesterol contents using commercial kits (Nanjing Jiancheng Bioengineering Institute, Nanjing, China). The endpoint was determined by measuring absorbance at 500 nm or 550nm using a microplate reader.

In summary, we have revised the manuscript in close conformity with the requests of the reviewers. We hope that the revised articles can be published in *Communications Biology*. We thank the reviewers for their excellent comments and we expect for your reply.

With best regards,

Sincerely yours,
Xiaoqing Wang, MD.
Xiaoqing_Wang@uestc.edu.cn

Dear reviewers,

On behalf of my colleagues and myself, I would like to submit a final revision of our manuscript entitled “**ABCC4 impairs the clearance of plasma LDL cholesterol through suppressing LDLR expression in the liver**” for consideration of publication in *Communications Biology*. The following are our responses to the comments of the reviewers.

We thank reviewers for their careful and intensive reading and thoughtful comments on our manuscript. We have carefully taken their comments into consideration in preparing our final revision, which has resulted in a paper that is clearer, more compelling. The following provides a summary of our responses to the reviewers' comments. Thanks for considering our manuscript.

Best wishes.

Responses to the comments of the reviewers

Reviewer #1 (Remarks to the Author):

The authors have clarified all the points raised by the Reviewer.

Answer: We thank the reviewer for the excellent comments which we believe can improve the final version of our manuscript. And, we have thoroughly checked the manuscript in close conformity with the requests of the reviewer.

Minor points.

1. Legend to Extended Data Fig.5a. I am not sure that a simple Student's test was applied for the testing the association between the SNPs and LDL-C levels. This point must be clarified if necessary.

Answer: We thank the reviewer for this insightful comment. We have confirmed that association analyses between SNPs and LDL-C levels in Extended Data Fig. 5a were conducted using a multivariable-adjusted linear regression model (adjusted for age and sex). The figure legend has been revised accordingly to clarify this.

2. Line 212: “chanegs” must be corrected and replaced by “changed”.

Once again, I think that the manuscript must be thoroughly checked before further editing process (few typos and spelling errors, check the numbering of Figures.....).

Answer: We sincerely thank the reviewer for pointing out the typo on Line 212, which has been corrected in the revised manuscript. In addition, we have carefully rechecked the entire manuscript for typographical errors, spelling mistakes, and any inconsistencies in figure and table numbering.

In summary, we have revised the manuscript in close conformity with the requests of the reviewers. We hope that the final revision of our manuscript can be published in *Communications Biology*. We thank the reviewers for their excellent comments and we expect for your reply.

With best regards,

Sincerely yours,

Xiaoqing Wang, MD.

Xiaoqing_Wang@uestc.edu.cn